# Impact of a candidate vaccine on the dynamics of salmon lice (*Lepeophtheirus salmonis*) infestation and immune response in Atlantic salmon (*Salmo salar* L.)

Jaya Kumari Swain[1][☯][¤]*, Yamila Carpio[2][☯]*, Lill-Heidi Johansen[1], Janet Velazquez[2], Liz Hernandez[2], Yeny Leal[2], Ajey Kumar[3], Mario Pablo Estrada[2]*

1 Nofima—The Food Research Institute, Tromsø, Norway, 2 Animal Biotechnology Division, Center for Genetic Engineering and Biotechnology, Havana, Cuba, 3 Symbiosis Centre for Information Technology, Symbiosis International (Deemed University), Pune, Maharashtra, India

☯ These authors contributed equally to this work.
¤ Current address: Fish Immunology and Vaccinology Research Group, Norwegian College of Fishery Science, UiT- The Arctic University of Norway, Tromsø, Norway
* jaya.k.swain@uit.no (JKS); yamila.carpio@cigb.edu.cu (YC); mario.pablo@cigb.edu.cu (MPE)

**Data Availability Statement:** All relevant data are within the manuscript and its Supporting Information files.

## Abstract

Infection with parasitic copepod salmon louse *Lepeophtheirus salmonis*, represents one of the most important limitations to sustainable Atlantic salmon (*Salmo salar* L.) farming today in the North Atlantic region. The parasite exerts negative impact on health, growth and welfare of farmed fish as well as impact on wild salmonid populations. It is therefore central to ensure continuous low level of salmon lice with the least possible handling of the salmon and drug use. To address this, vaccination is a cost-effective and environmentally friendly control approach. In this study, efficacy of a vaccine candidate, containing a peptide derived from ribosomal protein P0, was validated post infestation with *L. salmonis*, at the lab-scale. The sampling results showed good potential of the vaccine candidate when administered intraperitoneally in the host, in reducing the ectoparasite load, through reduction of adult female lice counts and fecundity and with greater presumptive effect in F1 lice generation. The sampling results correlated well with the differential modulation of pro-inflammatory, Th1, Th2 and T regulatory mediators at the transcript level at different lice stages. Overall, the results supports approximately 56% efficacy when administered by intraperitoneal injection. However, additional validation is necessary under large-scale laboratory trial for further application under field conditions.

## Introduction

Atlantic salmon (*Salmo salar* L.) is the most important economical species in aquaculture with a production value of 14.7 billion US dollars in 2014 [1] with Norway, Chile and Scotland being the top three salmon producers. However, with increased production the alarm about

**Funding:** This work was funded by Fiskeri - og havbruksnæringens forskningsfond, FHF (Norwegian Seafood Research Fund) grant number 901461(PI:JKS) (https://www.fhf.no/prosjekter/prosjektbasen/901461/). In addition, UIT- The Arctic University of Norway Open Access publication Fund covered the cost for publication. The funders had no role in the study design, data collection and analysis, decision to publish, or preparation of the manuscript.

**Competing interests:** The authors have declared that no competing interests exist.

the impact and number of diseases has also augmented, with parasitic salmon lice emerging as one of the most important in recent years in all the three major salmon-producing countries.

Two lice species represents primary concern for salmon farming: *Lepeophtheirus salmonis* in the Northern Hemisphere and *Caligus rogercresseyi* in the Southern Hemisphere [2]. In this study, we focused on, a single caligid copepod species *L. salmonis*, which predominates in the North Atlantic, causing year-round infestations of Atlantic salmon housed in marine cages, with concomitant ramifications for fish health in both farmed and wild salmonids as well as for aquaculture economics and sustainability [3]. However, the introduction of more and more salmon farms has significantly increased both the number and density of available susceptible hosts as well as parasite abundance in the coastal waters round the year [4].

Sea lice parasitize salmon during the marine phase of the life cycle, in both wild and farmed salmon, by attaching to their skin often close to gills and fins, feeding on the mucus, epithelial tissues and blood; reproducing on the host and releasing the eggs into the seawater. In seawater, the eggs hatch and develop into planktonic infective stages to parasitize the available host repeatedly [5], thus causing increased parasitic burden on the hosts. If left untreated, this can lead to impaired growth, osmoregulatory stress and open wounds, which can facilitate the entry of other pathogens [5, 6]. The impaired growth and secondary infections cause significant negative animal welfare and economic impact [7]. Moreover, relative to other salmonids, Atlantic salmon have limited ability to resist infection by *L. salmonis* and is therefore highly susceptible to the parasite [8]. The transfer of sea lice infestation from farmed to wild salmonids is of great concern [9]. Therefore, control of lice is the first basic priority for the industry, for further sustainable development. As a result, regulatory affairs departments in salmon producing countries have enforced strict limitations to the allowed sea lice levels in a farm. These regulations in turn impose treatments through different chemical, physical and biological methods at frequent intervals and thus directly increase the control-related costs.

However, pesticide use is significantly reduced now-a-days due to widespread resistance to these drugs and environmental pollution [10–12]. Increased frequency of treatment methods and salmon handling by drug-free treatments in the salmon farms has led to challenges with production cost, handling stress, injury, risk of secondary infection, mortality and thus impaired fish welfare. This has increased the necessity to develop new and alternative preventive measures [13, 14], which can document effect on lice and guarantee the fish welfare [15, 16]. To address this, vaccination against salmon lice could be an important alternative, since it is well-known that fish vaccines have greatly contributed to reducing the use of drugs (especially antibiotics) against several fish diseases.

Although *L. salmonis* has been an area of research for several decades [2, 3, 5, 6, 17], understanding the mechanisms behind the protection and development of prototype vaccines has been relatively slow and is still in its infancy. Approaches so far used have met with little or no success due to challenges in identification of protective antigens and mechanisms. Most strategies for sea lice vaccines have adopted similar approaches used for vaccines against other ectoparasites in mammals, for example vaccines against ticks [18].

The present study utilized a previously obtained vaccine candidate based on ribosomal protein P0 for its validation at the laboratory scale [19]. The P0 protein, having a molecular mass between 34–38 kDa, is highly conserved among eukaryotes [20]. The P0 peptide selected as antigen is located in a region of low sequence similarity between the lice P0 protein and those of its salmon host, in order to avoid the induction of tolerance in the parasite or production of auto-antibodies in the salmon host. In addition, to increase its immunogenicity, promiscuous T-cell epitopes (TCEs) from tetanus toxin and measles virus were fused to the N-terminus of a 35 amino acids peptide from the ribosomal P0 protein of *L. salmonis* [19]. These TCEs are universally immunogenic in mammalian immune systems [21] and reported to improve vaccine

efficacy in salmonids [22]. In our previous study, the candidate vaccine has shown to induce specific IgM response against pP0 compared to only synthetic pP0, in different teleost species including Atlantic salmon [19].

The purpose of this study was to investigate if the candidate vaccine is able to provide protection, either in terms of reduced lice count or reduced fecundity or both. Therefore, we targeted to study the impact of the candidate vaccine at different stages of parasite infestation, post immunization, under controlled laboratory conditions. Moreover, to highlight the vaccine's further impact on F1 generation hatching efficiency, egg strings collected from the parasitized adult female lice were hatched and compared. Simultaneously, host-lice interaction studies at the gene level were performed to explore the immune modulation in response to vaccination for the first time at different life stages of lice infestation.

## Materials and methods

### Antigen purification

The antigen is P0 based protein, which was granted patent for vaccine antigen (Vaccine composition for controlling ectoparasite infestations PCT/CU2011/000005). P0 based antigen protein was purified as described previously by Leal et al. (2019) [19]. Briefly, inclusion bodies were obtained by harvesting induced bacteria cells and centrifugation at 10,000 x g for 10 min at 4°C. The cell pellets were resuspended in 300 mM NaCl, 10 mM Tris, pH 6 and were disrupted in French Press (Ohtake, Japan) at 1 200 kgf/cm$^2$. The disrupted cell suspension was centrifuged at 10,000 x g for 10 min at 4°C and the cell pellet containing the protein was suspended in 1M NaCl, 1% Triton X-100 using politron Ultra-Turrax T25, IKA WERKE and centrifuge again at 10,000 x g for 10 min at 4°C. This step was repeated once again and purified inclusion bodies were suspended in PBS (16 mM $Na_2HPO_4$, 4 mM $NaH_2PO_4$, 120 mM NaCl, pH 7.4). Protein concentration was determined by bicinchoninic acid assay (BCA) assay (Pierce, USA) according to the manufacturer's instructions and by densitometry scanning of protein gels. Protein samples were checked by SDS-PAGE on 15% polyacrylamide gels and western blotting.

### Fish husbandry

The experiment was approved by the Norwegian Food Safety Authority, (Mattilsynet), application ID 14617 (https://www.mattilsynet.no/sok/?search=ID+14617). The oversight of the animal welfare and care was undertaken as a part of the approval from the Norwegian Food Safety Authority. In Norway, the National Animal Research Authority (NARA) equivalent to animal ethics committee is a part of Norwegian Food Safety Authority (Mattilsynet). The experiment was performed at the Aquaculture Research Station (Tromsø, Norway). Atlantic salmon (AquaGen standard, average weight 40 ± 6), at a density of 10 kg/m$^3$ were kept in circular 500 L tanks supplied with filtered circulating fresh water for 2 weeks at an ambient temperature of approximately 10°C with 24 h light (summer stimuli) for acclimation. Fish were fed with a commercial pellet diet (Nutra Olympic, Skretting).

### Fish immunization

Three 500 L tanks were stocked with 120 fish each, one tank assigned to each of the 3 experimental groups: procedural control (Group 1), injected vaccine (Group 2), and injected vaccine + bath immunization (Group 3). Each tank was supplied with continuous circulating water flow throughout the experimental period and oxygen level and temperature were recorded daily. For vaccine formulation, recombinant antigen protein or PBS control were adjuvanted

in Montanide ISA50 V2 (Seppic, France) at a ratio of 50/50. Montanide ISA50 V2 has been successfully used in salmon as an adjuvant [23]. Our previous observation has also shown that ISA50 works better compared to fish specific adjuvant ISA763 (unpublished observation).

Immunization and challenge schedule are outlined in Fig 1. The fish were starved for one day before vaccination. Prior to vaccination, fish were anaesthetized in 0.005% benzocaine. First immunization was performed as follows: each fish in the control group (Group 1) received 0.05 mL of PBS emulsified in adjuvant by intraperitoneal (ip) injection; second group (Group 2) received ip injection at a dose of 1 μg/gram body weight (gbw) of the recombinant antigen protein emulsified in adjuvant; and third group (Group 3), received ip injection at 1 μg/gbw of the recombinant antigen protein emulsified in adjuvant plus bath immunized with recombinant antigen protein as inclusion bodies (200 μg/L) for 1 hour (120 fish in 200 L aerated static bath), immediately after ip injection.

Fifteen days post immunization, fish were transferred to seawater. After 37 days post first immunization, a booster dose was given to each fish (average weight 60 ± 10 g) at a similar dose per gram of body weight as first vaccination. Each experimental group was then kept in duplicate in 300L tanks with a stocking density of approximately 10 kg/m$^3$. Throughout the experiment, the following experimental conditions were maintained: Temperature: 10˚C; Light: 24 h; Oxygen level at outlet: ~80–90%; Salinity: 34–35 ppt.

## *In vivo* lice challenge

After one month of booster dose, 90 fish (average weight 94 ± 16 g) from each group were bath challenged with infective copepodids of *L. salmonis* (Oslo/Gulen strain from Norwegian Institute of Marine Research, IMR). The groups were bath challenged in separate 500 L tanks for one hour with stopped water supply keeping the oxygen level stable through aeration. Each 500L tank received approximately 3150 copepodids to have an average distribution of about 35

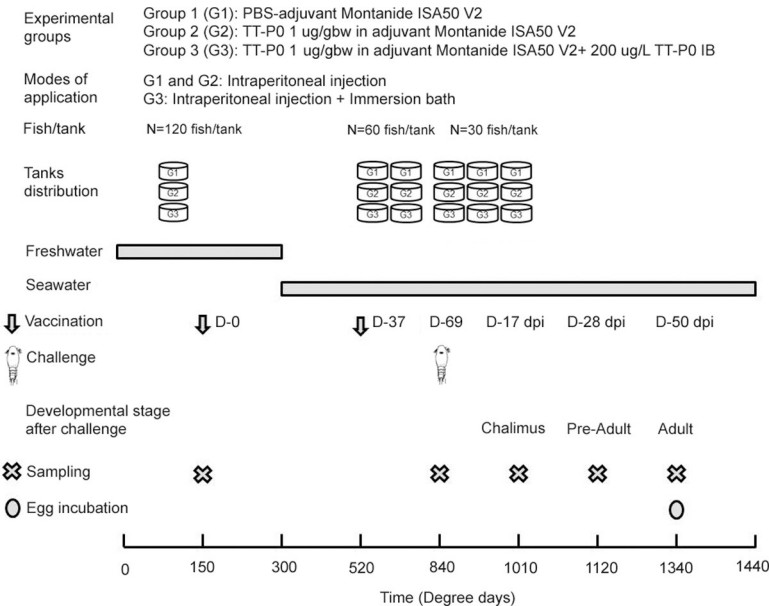

**Fig 1. Experimental outline.** Experimental design depicting experimental groups, immunization, challenge, post challenge schedule along with sampling time-points. Group details: Group 1 is control group; Group 2 received ip injection of adjuvant emulsified vaccine antigen TT-P0; Group 3 received ip injection of adjuvant emulsified vaccine antigen TT-P0 + bath immunization with TT-P0 inclusion bodies (IB).

copepodids/fish. Post challenge each group was distributed into triplicate tanks (500L) with 30 fish per tank and a flow rate of 5 L/min. The parasitized fish were kept in seawater with a salinity of 34.5‰, oxygen level: 80–90% and at a temperature of approximately 10˚C, until the salmon lice reached desired developmental stage i.e at matured adult stage when females have developed egg strings.

## Sampling and lice counting

To evaluate at which developmental life stage of lice the vaccine candidate was effective, counting of lice on 10 parasitized fish per tank i.e. 30 fish per group, were performed at 17 days post infestation (dpi) (chalimus), 28 dpi (pre-adult) and 50 dpi at mature adult stage with first reproductive egg strings (Fig 1). At 50 dpi i.e at the adult stage, both egg string number and egg string length per female were noted, for all the immunized groups and compared. Fish were taken out one at a time gently by hand net from the stocking tanks and transferred in a bucket containing a lethal dose of anesthetic water (0.01% benzocaine). Care was taken for the minimum loss of lice through netting. All fish were treated the same and were handled gently. The hand nets were checked for detached lice and the net mesh was fine enough to capture lice if any lice would have fell off. However, we cannot exclude the possibility that some egg strings or lice would have been detached from the fish or lost in the tank during handling (netting). One fish per bucket was anaesthetized before lice counting and sampling of tissues. To avoid counting error of detached lice due to anesthesia and handling, counting of chalimus at 17 dpi, pre-adults at 28 dpi and adults at 50 dpi on individual parasitized fish were performed under water in a white tray. After lice counting from each fish per tray the remaining water in the tray and bucket were checked for detached lice.

At pre-challenge sampling points, fish were humanely euthanized using 0.01% benzocaine prior to measurement of length and weight as well as collection of different tissues (skin, spleen and head kidney) for gene expression study. Further, spleen, head kidney and skin tissues were sampled aseptically from 18 fish per group (6 fish/tank). Tissue samples were immediately transferred to RNA later (Ambion) and kept at 4˚C overnight and then stored at -20˚C. Overall sampling time-points as outlined in Fig 1 were at 0 (prior to 1st vaccination), 69 (31 days post booster) days post vaccination or 0 day challenge, and 17 (chalimus), 28 (pre-adult) and 50 (adult) dpi.

**Incubation of collected egg strings for F1 generation hatching.** To analyze the effect of vaccine on F1 generation copepodids production, first reproductive egg strings, detached from gravid females at 50 dpi were incubated in well-aerated filtered seawater. This was to determine the effect, vaccine candidate had on hatching efficiency of the F1 generation copepodids. Fifty egg strings (sampled from the first reproductive event at 50 dpi) from each experimental group were randomly distributed and incubated in 5 parallel aerated flow-through incubators (containing 500 mL filtered seawater/incubator at ~10˚C) having 10 egg strings per incubator for 8 days, to study the hatching success to F1 generation copepodids. First visual observation was done on day eight post incubation and final counting was performed at day ten. Copepodids density was estimated by taking 10 mL water samples from each replicate and counting of copepodids was performed using dissecting microscope. This observation was repeated four times for each replicate.

## Vaccine efficacy

The overall efficacy (in percentage) of the candidate vaccine was calculated using the following formula:

$$\text{Vaccine efficacy } (\%) = 100 \text{ x } [1 - (\text{FE x LE x CE})]$$

Where FE is the effect on female survival to maturity, LE is the effect on fecundity of adult females (using egg string length as a proxy for fecundity), and CE is the effect on hatching and survival of F1 offspring to the copepodid stage.

## Observation of side effects

Usually, adjuvanted vaccines may cause inflammation, granuloma and pigmentation at the site of injection. To check the side effects of the candidate vaccine having the Montanide ISA50 as an adjuvant, visual scoring and analysis of the vaccine side effects resulting in adhesion was performed using the Speilberg scoring method according to the criteria detailed by Midtlyng et al. (1996) [24]. A separate score for pigmentation for each fish was assigned according to the table in Fig 4B. Fish weight and length were registered and the condition factor (K) was calculated according to Barnham and Baxter (1998) [25] using Fulton mathematical formula:

$$K = (10^N W)/L^3$$

K: is the Condition Factor or Coefficient of Condition; often referred to as the "K factor".
W: is the weight of the fish in grams (g).
L: is the length of the fish in millimeters (mm). In the case of salmonids, length is measured from the tip of the snout to the rear edge of the fork at the center of the tail fin; known as length to caudal fork (LCF). The cube of the length is used because growth in weight of salmonids is proportional to growth in volume.
N = 5; having weighed and measured thousands of salmonids from Victorian waters, the value of N used by the Department for determining K is set at this figure to bring the value of K close to unity.

## Gene expression studies

All organs from the sampled fish, kept in RNA-later (Ambion, Austin, TX, USA) were subsequently processed for RNA isolation. Total RNA was extracted by MagMAX™-96 Total RNA Isolation Kit (Invitrogen), including turbo DNase treatment (Invitrogen) according to manufacturer's instruction.

Analysis of gene expression by Real-time PCR (QPCR) was performed in duplicates with a QuantStudio 5 Real-Time PCR System (Applied Biosystems) using SYBR Green (Applied Biosystems) in 384 well plates. For each mRNA, gene expression was normalized to the geometric mean of the 3 house-keeping genes (EF-1a, 18S and β-actin) in each sample and fold change was calculated according to Pfaffl method [26] considering the primer efficiency (E). Primer sequences used for gene expression studies are listed in Table 1.

## Statistical analysis

The results were analyzed and expressed as mean ± standard deviation (SD) unless otherwise stated. SD was calculated across all fish within a group. Statistical analysis was performed and graphs were made using the Prism 6.01 software for Windows (GraphPad software, San Diego, CA, USA). For lice counts and fecundity parameters, Mann-Whitney test was performed due to unequal variances to compare vaccinated groups (Group 2 or 3) with control (Group 1). Prior to individual gene expression data analysis, outliers were calculated and identified using the ROUT method through Prism 6.01 software for Windows and were removed from the subsequent gene expression statistical analysis. Normal distribution was assessed using D'Agostino & Pearson omnibus normality test. Multiple comparison were performed using analysis of variance (ANOVA) or Kruskal Wallis test depending on the normal distribution and equal

**Table 1. Primer sequences used for the real-time PCR analysis.**

| GENE TARGET | NAME | ACCESSION No. | FORWARD (5–3') | REVERSE (5–3') | AMPLI-CON |
|---|---|---|---|---|---|
| Immuno-globulin M (secretory) | *IgMs* | BT060420 | CTACAAGAGGGAGACCGGAG | AGGGTCACCGTATTATCACTAGTTT | 90 |
| Immuno-globulin T | *IgT* | GQ907004 | CAACACTGACTGGAACAACAAGGT | CGTCAGCGGTTCTGTTTTGGA | 97 |
| Tumor necrosis factor alpha1 | *TNFα1* | AY929385 | ACTGGCAACGATGCAGGACAA | GCGGTAAGATTAGGATTGTATTCACCCTCT | 144 |
| Interleukin 1 beta | *IL-1β* | AY617117 | GCTGGAGAGTGCTGTGGAAGAAC | CGTAGACAGGTTCAAATGCACTTTGTG | 220 |
| Interferon gamma | *IFN-γ* | AY795563 | GATGGGCTGGATGACTTTAGGATG | CCTCCGCTCACTGTCCTCAAA | 166 |
| Interleukin-4/13A | *IL-4/13A* | EG837625 | CCACCACAAAATGCAAGGAGTTCT | CCTGGTTGTCTTGGCTCTTCAC | 147 |
| Cluster of Differentiation 4 | CD4 | EU585750 | CGGAAGCGAGGGATATAAATGGTG | GGCATCATCACCCGCTGTCT | 215 |
| Cluster of Differentiation 8 alpha | CD8α | AY693393 | GACAACAACAACCACCACGACTACAC | GCATCGTTTCGTTCTTATCCGGTT | 211 |
| Matrix metallo-proteinase-9 | *MMP-9* | AGKD01108865 | TGGAGAGAACTACTGGAGGCTGGA | CCGACAGAAGTAGATGTGGCCCTT | 142 |
| Interleukin 8 | *IL-8* | HM162835 | TCCTGACCATTACTGAGGGGATGA | AGCGCTGACATCCAGACAAATCTC | 200 |
| Interleukin 10 | *IL-10* | EF165028 | CTGTTGGACGAAGGCATTCTAC | GTGGTTGTTCTGCGTTCTGTTG | 129 |
| Interleukin 22 | *IL-22* | DW572073 | GGCCCGAGTCAGCAGAGACCT | CTCCTCCATCCCGGCCAACTTC | 106 |
| Beta actin* | *β-actin* | BT059604 | CAGCCCTCCTTCCTCGGTAT | CGTCACACTTCATGATGGAGTTG | 72 |
| Elongation factor 1-α* | *EF1α* | AF498320 | CAAGGATATCCGTCGTGGCA | ACAGCGAAACGACCAAGAGG | 327 |
| 18 S ribosomal RNA* | 18 S rRNA | AJ427629 | TGTGCCGCTAGAGGTGAAATT | CGAACCTCCGACTTTCGTTCT | 101 |

(*) indicates reference genes used in this study for normalization.

variance of the data followed by Tukey or Dunn's Multiple Comparison *post hoc* tests. P-values < 0.05 were considered statistically significant. Two-way hierarchical clustering analysis heat map and dendrogram of relative gene expression data and experimental groups were generated in R language using ComplexHeatmap package by Gu, Z et al. (2016) [27]. For Principal component analyses, "FactoMineR" package of the R statistical software (v3.6.2) was used to calculate the principal components and visualizations were constructed using "factoextra" package. Ellipses in the PCA graph are confidence ellipses with a confidence level of 0.95 and the centroids represent the center of the mass of the points per group.

## Results

### Impact of the vaccine candidate post lice infestation

At 17 dpi, mean number of chalimus (± SD) attached per fish was 20.00 (± 8.08), 25.17 (± 10.02) and 23.70 (± 12.41) for group 1, group 2 and group 3, respectively (Fig 2A). No significant differences among groups were detected at 17 dpi. At 28 dpi, mean PA count per fish was 12.83 (± 6.29), 12.73 (± 5.13) and 17.77 (± 7.28) for control, group 2 and 3, respectively, where group 3 showed more PA per fish as compared to groups 1 and 2 (*P* < 0.01). Finally, at 50 dpi, mean infestation rate of adult lice per fish was reduced to 5.13 (± 2.94), 4.06 (± 2.53) and 5.50 (± 2.63) for group 1, 2 and 3, respectively, and compared to control, group 2 showed an overall reduction tendency of 21% (Fig 2A and Table 2), although not significant. Moreover, development rates of *L. salmonis* throughout the experiment were nearly identical between the immunized and the control group.

Although there were no great differences between the total lice counts per fish on the immunized and control fish regardless of louse life stages, statistically fewer adult female lice (40% reduction), and fewer number of female lice with eggs (42.5% reduction) per fish were present on group 2 immunized fish compared to control. However, no differences were

observed in group 3 compared to control (Fig 2B and 2D and Table 2). During the 50 dpi sampling, all egg strings were collected from the gravid females. Most of the gravid females had two egg strings. Gravid female lice removed from the immunized fish showed shorter egg string length compared to the control group, of which group 3 had significant reduction of 6% ($P < 0.04$) (Fig 2E and Table 2). The results mentioned above clearly showed reduced number of eggs produced by females in group 2 (42.5% reduction, $P < 0.04$) and thus supports significantly reduced fecundity in terms of reduced egg string data and less gravid females in group 2 (Table 2). Overall, lice-induced damage on the parasitized fish was low and no wounds were visually observed on any of the experimental fish. Furthermore, there was no evidence of any secondary infections either on the surface or in internal organs of the infected fish.

## Impact on hatching efficiency of F1 generation copepodids

Post egg string measurement, a total of 50 egg strings from each experimental group were divided into 5 replicates (10 egg strings per replicate), for the F1 incubation experiment. Fig 2F shows the leftover egg strings (from the total egg strings collected from gravid females at 50 dpi), after the removal of 50 egg strings for the hatching experiment. During incubation, hatching of the egg strings were followed in each group to check if the reduced female fecundity of group 2 in F0 generation had any consequences in the early F1 generation. Subsequently, the F1 copepodids were observed on day 8 and counted on day 10 post-incubation, and data were analyzed. At day 8, the hatching success of egg strings removed from lice on the immunized group were delayed and reduced, especially in group 2, compared to the control

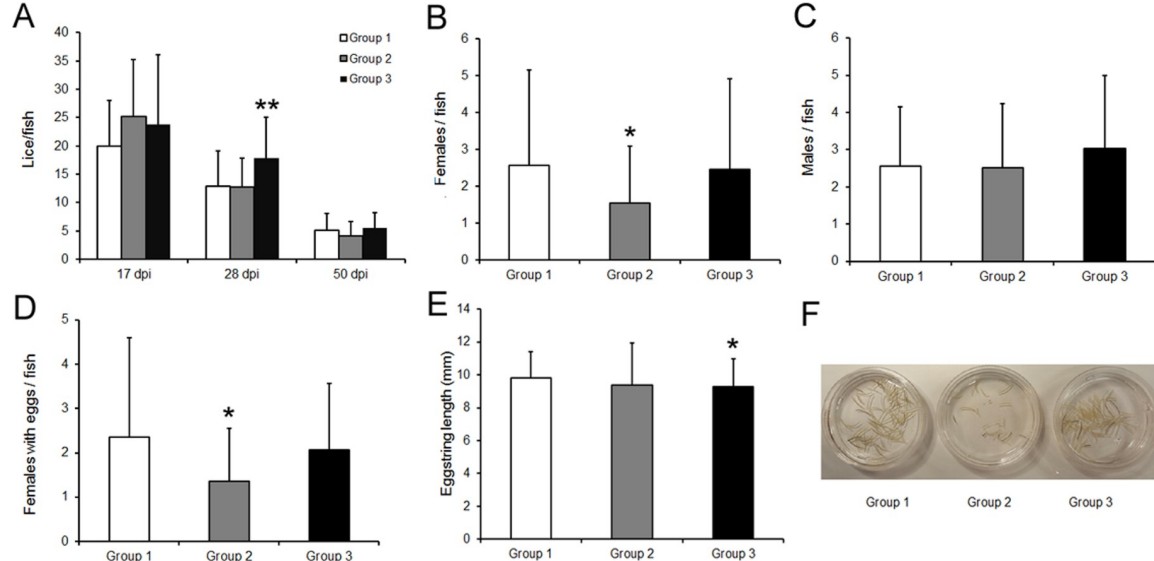

**Fig 2. Efficacy of TT-P0 vaccine on lice counts and fecundity of adult female lice post infestation.** (A) Bar graph showing average lice count / fish for different immunized groups at different lice stages post infestation (dpi): chalimus (17 dpi), pre-adult (28 dpi) and adults (50 dpi). Adult lice on the experimental fish were sampled at 50 days post infestation (dpi). The sampled lice were counted for total number of males, females and female's fecundity parameters per fish. Data showing, (B) Female numbers, (C) Male numbers, (D) Females with eggs, (E) Egg string length, for different groups per fish at 50 dpi. Data shown as mean + SD. A Mann-Whitney test was performed due to unequal variances to compare vaccinated groups (Group 2 or 3) with control (Group 1). Asterisk indicates statistically significant differences compared to control group ($^*P < 0.05$, $^{**}P < 0.01$). (F) Photograph of leftover egg strings (after removal of 50 egg strings for F1 generation hatching experiment) representing the visual number of total egg strings in different groups. Group details: Group 1 is control group; Group 2 received ip injection of adjuvant emulsified vaccine antigen; Group 3 received ip injection of adjuvant emulsified vaccine antigen + bath immunization. Sampling was done from 10 fish per tank, 3 replicate tanks, thus 30 fish per experimental group.

**Table 2. Effect of vaccination on salmon lice infestation following adult stage of lice.**

| Experimental groups | Number of Fish (n) | Reduction of adult lice number | Reduction of adult females | Reduction of gravid females with eggs | Reduction of egg string length (mm) | Reduction in F1 copepodids |
|---|---|---|---|---|---|---|
| Group 2 | 30 | 21% | 40% * | 42.5% * | 5% | 23% |
| | | | (P < 0.02) | (P < 0.04) | | |
| Group 3 | 30 | 3% | 8% | 22% | 6%* | 4% |
| | | | | | (P < 0.04) | |

*Shows significant difference with respect to control group. Group 2 is ip vaccinated group; Group 3 is ip vaccinated + bath immunization group. The percentage values were derived compared to control (group 1) from the data shown in Figs 2 and 3.

group (S1 Table). This correlates well with the reduced fecundity in the F0 generation of group 2 gravid females. However, the counting at day 10 showed a reduction of 23% and 4% infective copepodids in the vaccinated group 2 and 3, respectively (Fig 3 and Table 2). The percentage reduction of copepodids on day 10 was not high, as expected based on observation made on day 8 (S1 Table). This was due to some unseen or technical problem occurred during the week-end, resulting in some unexpected mortality of the copepodids before counting on day 10. The experiment was not possible to repeat due to limited time and resources available.

Overall, the vaccine efficacy of group 2 was the best among the groups with an efficacy of 56%, whereas group 3 efficacy was 25%. The terminology "vaccine efficacy" used here should not be interpreted as protection obtained. The proposed formula aims to evaluate the impact of vaccination over female lice fecundity and the effect on hatching and survival of F1 offspring to the copepodid stage.

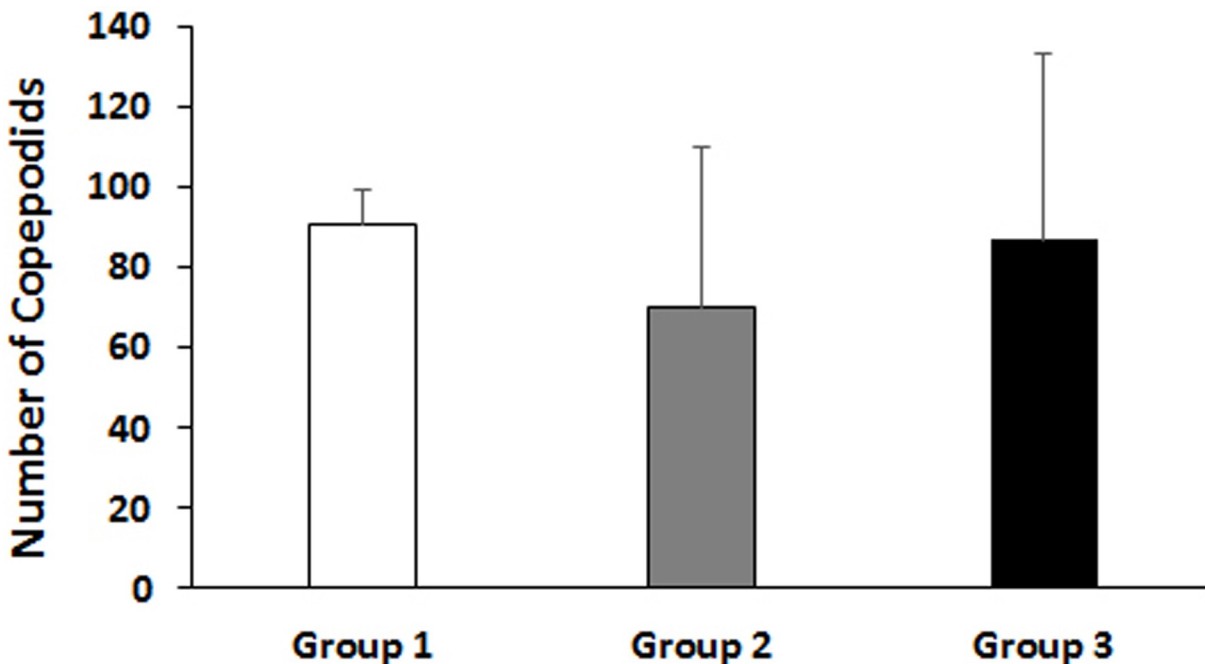

**Fig 3. TT-P0 vaccine's effects on F1 generation hatching and copepodids number.** Bar graph showing total number of copepodids 10 days post incubation of egg strings. Fifty egg strings (sampled from the first reproductive event at 50 dpi) from each experimental group were randomly distributed and incubated in 5 parallel aerated flow-through incubators (containing 500 mL filtered seawater/incubator at ~10°C) having 10 egg strings in each incubator. The bar shows the mean value + SD in 5 replicate incubators for each experimental group. Group details: Group 1 is control group; Group 2 received ip injection of adjuvant emulsified vaccine antigen; Group 3 received ip injection of adjuvant emulsified vaccine antigen + bath immunization.

## Side effects analysis

Fish weight, length and condition factor (K) were analyzed at all sampling points. Immunized fish had less weight and length post lice challenge as compared to the control group (Fig 4A). The reduction in weight in groups 2 and 3 compared to group 1 were 13–10%, respectively. The reduction in length were 6% in both the vaccinated groups compared to control group 1. The condition factor was acceptable (1.2) and it was the same for all the groups at different sampling times (Fig 4A). For salmonids, K values usually fall in the range 0.8 to 2.0 [25].

Moreover, side effects of the candidate vaccine having the Montanide ISA50 V2 adjuvant were analyzed using Speilberg and pigmentation scoring at 50 dpi. Speilberg scoring at 50 dpi showed that the control group had an average score of 2.0 compared to group 2 and 3, which showed an average score of 2.8, i.e below 3, which is in an acceptable range for adhesion (Fig 4B).

Pigmentation score was significantly less in the immunized groups compared to the control group, as shown in Fig 4B. Pigmentation was observed only on the epithelial lining and not in muscle or tissue within the peritoneum.

Simultaneously, individual fish checked for vaccine depots had vaccine residues, which were encapsulated by connective tissue as small pockets. The injection site was checked for redness and lesions and looked normal in all the fish.

## Effect of vaccination combined with *L. salmonis* infestation on tissue specific gene expression

Gene expression of pro-inflammatory mediators (TNF-α, IL-1β, IL-8); Th17 and regulatory mediators (IL-22, IL-10); Th1 and Th2 mediators (IFN-γ, IL-4/13A); immunoglobulin genes and cellular markers (IgM, IgT, CD4, CD8α); and tissue remodeling gene matrix metalloprotease 9 (MMP9), were studied to evaluate the immune response of vaccinated fish to salmon

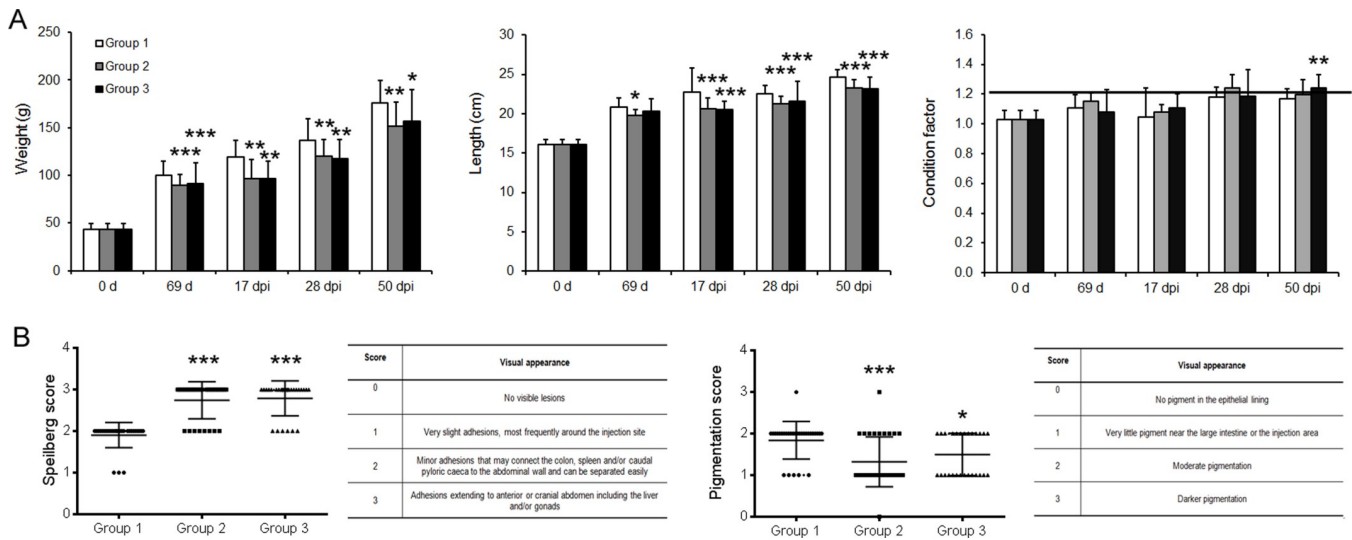

**Fig 4. Side effects of TT-P0 vaccine post immunization.** (A) Growth (weight and length) and condition factor of the fish post immunization and challenge at different sampling points: Pre-immunization (0 d), 69d post immunization (69 d) and at different days post infestation (dpi) based on the different lice stages during infection: 17 dpi (chalimus), 28 dpi (pre-adult) and 50 dpi (adult). (B) Visual scoring and analysis of the vaccine side effects resulting in adhesion (left panel) and pigmentation (right panel) near the vaccination site. Data are shown as the mean + SD of the parameters under analysis (n = 30). Based on normal distribution test, one-way ANOVA or Kruskal Wallis test was done followed by Tukey or Dunn's Multiple Comparison. Asterisk (*) indicates statistical difference *(P < 0.05), **(P < 0.01), ***(P < 0.001) between the groups with respect to control (Group1). Group details: Group 1 is control group; Group2 received ip injection of adjuvant emulsified vaccine antigen; Group 3 received ip injection of adjuvant emulsified vaccine antigen + bath immunization.

lice infestation at different stages of their life-cycle, compared to control fish. Both anterior kidney and spleen, the main immune organs in teleost fish, were used to evaluate systemic responses and skin was used to evaluate the local immune response to salmon lice infestation.

**Global assessment: Heat map and hierarchical clustering.** To obtain an overview of the expression profiles of the different groups tested at different sampling points corresponding to different lice stages, heat maps were constructed with hierarchical clustering. Hierarchical clustering of all the genes studied, identified three clusters representing a differential clustered expression pattern with respect to spleen tissue (Fig 5A).Hierarchical clustering of the experimental groups at different sampling time points pre and post infestation (Fig 5B), also identified 3 clusters for all the tissues studied, showing differences in gene expression under different lice infestation stages and treatment groups. Gene expression cluster comparison showed that the pro-inflammatory cytokines, T-regulatory mediators, Th1 and Th2 mediators and T cell surface markers were strongly clustered. A clear pattern of different upregulated gene clusters were visible in different tissues, showing highly upregulated cluster of pro-inflammatory cytokines genes in spleen, highly upregulated regulatory cytokine genes in head kidney and mixed upregulated gene expression of Th1, Th2, T reg, IgM and IL-8 in skin. These results showed that, apart from lymphoid organs, local response played a major role during the host-parasite interaction at later stages post infestation i.e 28 dpi in the vaccinated groups (group 2 and 3). On the other hand, column-wise comparison based on different sampling time-points, within respective groups, showed strong clusters with respect to substantial gene upregulation at 28 dpi in vaccinated groups (group 2 and 3) in skin, at 17 and 50 dpi (group 3), and 28 dpi (group 2) in spleen and at 17 dpi (group 2 and 3) as well as 28 dpi (group 2) in head kidney. Consequently, evaluating the two-way hierarchical clustering analysis for all the tissues, vaccinated group 2 at 28 dpi showed the highest number of upregulated genes compared to the control group. However, vaccinated group 3 showed higher number of upregulated genes at 17 dpi in spleen and head kidney and at 28 dpi in skin. Heat map with two-way clustering of genes studied in the individual tissue is given in Figs 1 in S1.

**Principal component analysis (PCA).** We performed exploratory data analyses using principal component analysis (PCA) in all the tissues studied. The PCA analysis of the expression profile of the 12 selected genes in skin samples at different time points post lice infestation (Fig 6) showed that samples taken at the early stages of infection [0 day challenge (69 d) and 17 dpi] in vaccinated and control groups were very similar and with low variability. Consecutive samples (28 and 50 dpi) displayed an increasing deviation along the principal component 1 (PC1) that contributed to most (78.5%) of the observed variation. Samples taken at 28 and 50 dpi formed clearly distinct clusters, and variability among individual sampling points within groups increased with infestation time. Moreover, 28 dpi in vaccinated group 2 contributes to maximum percentage variation (~43%) in PC1 (Fig 6A and 6C). All the 12 genes studied showed significant ($P<0.05$) contribution in PC1 (Fig 6C) and in addition, IgT expression showed significant contribution in PC2 where 50 dpi in group 3 had maximum contribution. For head kidney and spleen, PC1 component contributed to 67.9 and 64.3% variation, respectively at 17 and 28 dpi in both the vaccinated groups (S2 and S3 Figs). Similar to skin, in head kidney also 28 dpi in vaccinated group 2 contributes to maximum percentage variation (~37%) in PC1. On the contrary, in spleen 17 dpi vaccinated group 2 showed maximum contribution of ~34% followed by 28 dpi from vaccinated group 2 (~19%) (S2C and S3C Figs). All genes showed significant contribution in PC1 expect for IL-10 and MMP9 in head kidney and CD8α in spleen (S2D and S3D Figs). This showed that vaccination together with lice infestation has significant effect on the overall gene expression profile with more significant contribution at 28 dpi in group 2 than group 3.

**A**

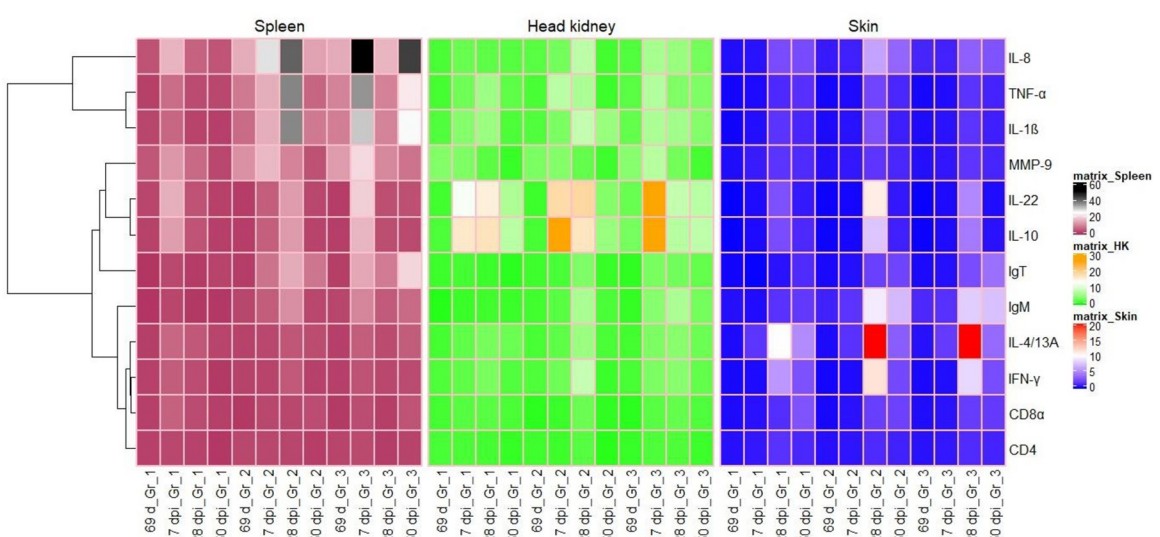

**B**

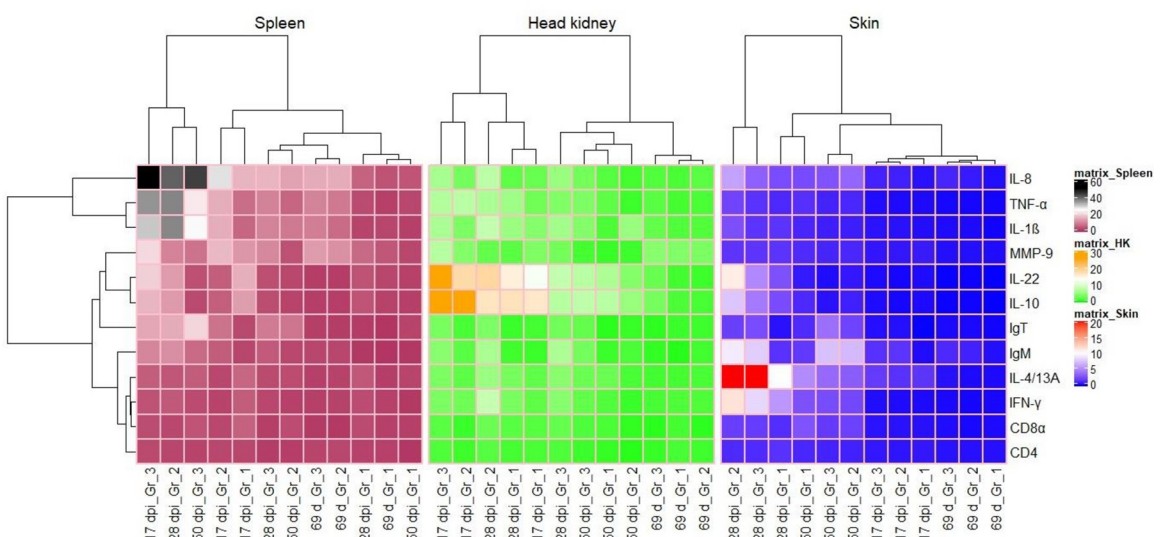

**Fig 5. Hierarchical clustering analysis heat map and dendrogram of relative gene expression data over different sampling time points pre and post infection within the vaccinated groups, and for three different tissues.** (A) indicates the pattern of gene expression across different groups and tissues. It also shows one-way clustering of differentially expressed genes on the right with respect to spleen whereas (B) shows two way hierarchical clustering of genes on the right and group wise sampling time-points on the top. Differential gene expression is represented for all genes as a colour gradient across all sampling points within different groups from brick red (lowest) to black (highest) for spleen, green (lowest) to dark orange (highest) for head kidney, blue (lowest) to red (highest) for skin. Group details: Group 1 is control group; Group2 received ip injection of adjuvant emulsified vaccine antigen; Group 3 received ip injection of adjuvant emulsified vaccine antigen + bath immunization.

**Detailed assessment by individual gene expression analysis.** The results from the overview of gene expression profiles and the exploratory data analyses clearly showed changes

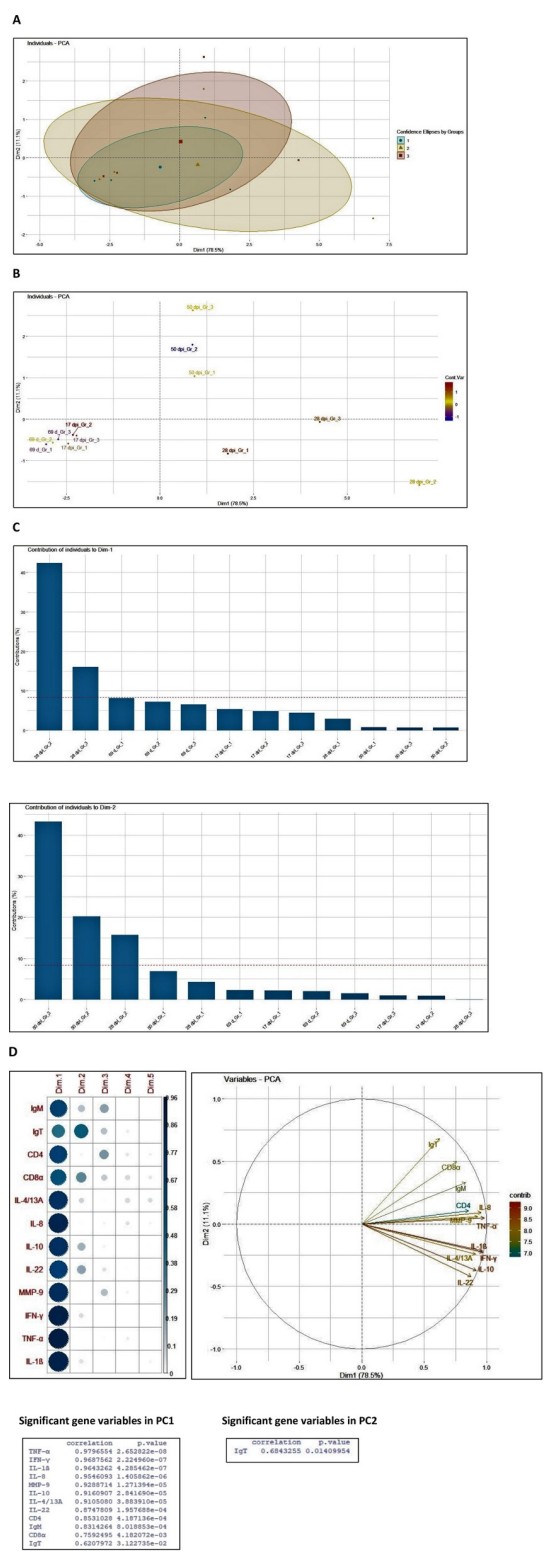

**Fig 6. PCA analysis of skin samples pre and post lice challenge.** PCA analysis for the *in vivo* challenge samples representing the distribution of lice infested host skin samples in vaccinated (group 2 and 3) and only adjuvant vaccinated control (group 1) groups at 0, 17, 28 and 50 dpi (A and B). Analysis was based on mean fold-changes of all genes for each individual sample at each sampling point (smaller symbols) relative to the unvaccinated control. The ellipses indicate the group dispersion/variability from the centroid (larger symbols) calculated using all individual fold-

changes values/group (A). (C) shows the contribution of sampling points to different components. (D) shows the contribution of genes on different components and the significant genes contributing in principal component 1 and 2.

related to different sampling points post lice infestation and vaccination groups. Therefore, we proceeded to study these changes in detail, to further characterize gene expression levels. The overview of the relative gene expression of all the genes analyzed in this study is graphically represented in S4 Fig. The gene expression results in spleen (S4A Fig), showed that the pro-inflammatory cytokines TNF-α, IL-1β and chemokine IL-8 were significantly up-regulated starting from pre-challenge (69 d) and this trend was maintained until 50 dpi in the immunized groups compared to the control. The same results were obtained for metalloproteinase 9 (MMP-9) except at 28 dpi in spleen (S4A Fig). For IL-1β, there was also significant expression in head kidney across all sampling time-points except 17 dpi in both vaccinated groups. On the other hand, in skin IL-8 expression was up-regulated only at 28 dpi in group 2 and at both 17 dpi and 28 dpi in group 3 (S4A Fig). However, IL-1β and TNF-α in skin was downregulated in infected salmon at 50 dpi in both the vaccinated group, respectively compared to control except at day 69 (pre-challenge), where TNF-α was significantly upregulated in group 3.

The gene expression results also showed that both IgM and IgT transcript levels were significantly upregulated in the vaccinated groups compared to control in all the tissues and sampling time-points studied (S4B Fig) with exception of IgT in head kidney at 69 day post vaccination and 17 dpi. Both genes followed almost similar pattern of expression in different groups and sampling time-points, suggesting its important role in host-parasite interaction. IgT transcription in skin was up-regulated earlier, at 69d (0 day infestation) in group 3.

On the other hand, the activation of T-cell related genes: CD4, CD8α, IL-4/13A and IFN-γ showed significantly higher expression levels in the spleen at 50 dpi (S4B Fig). This trend was also seen at 28 dpi in the head kidney showing the activation of T-cell mediated immunity and the involvement of Th1/Th2 response. Significant decreasing trends of expression levels were also found in these genes at other sampling points showing different patterns of regulation depending on sampling time or experimental groups. For example, cytotoxic T cell marker, CD8α transcript was downregulated compared to the control group in most of the sampling points other than the time-points mentioned above.

In addition, T-cell mediated Th17 and regulatory cytokines, IL-22 and IL-10 shared a common trend of gene expression (S4B Fig) without any specific significant up-regulation except group 3 at 50 dpi in spleen. They were significantly downregulated in group 2 fish at 17 dpi in spleen and in head kidney at 28 dpi, whereas in group 3 fish at 28 dpi only in spleen.

## Discussion

The importance of Atlantic salmon in aquaculture and its susceptibility to infestation with *L. salmonis* has led researchers to investigate efficient non-medicinal, cost effective and eco-friendly measures to control the sea-lice load through the possibility of vaccine development. In the current study, we used a vaccine based on the peptide of 35 amino acids from the ribosomal P0 protein of *L. salmonis* fused to the C-terminal of TCE's from tetanus toxin and measles virus positioned in tandem and previously tested for better antibody response [19]. Normally, housekeeping proteins are highly conserved among species and the development of a vaccine candidate based on housekeeping proteins such as P0 ribosomal protein is very challenging due to its high degree of identity between the P0 sequence of the vertebrate host and the ectoparasite. Consequently, the peptide P0 used as a vaccine candidate in this study was selected from the less conserved region between the *L. salmonis* and salmon.

According to the sampling results in this study, initially an overall average of about 23 attached lice at the chalimus stage were recorded from each fish sampled at 17 dpi and by the end of the experiment, these numbers reduced to about 5 adult lice per fish. The total number of lice attached at a particular developmental stage post infestation did not vary statistically between the immunized or control groups of fish, although there was a tendency of reduction at the adult lice stage in ip vaccinated group 2. However, significant impact on gravid female lice count and its reproductive efficacy with delayed hatching and reduced trend of copepodids count in F1 generation was documented also in group 2 compared to control group. This showed that the major effect of vaccine was apparent in the adult female lice and its fecundity. A similar impact on female's *R. B. microplus* population was seen after challenge when a 20 aa P0 peptide derived from *Rhipicephalus* ticks conjugated to KLH was used to immunize cattle [28]. They reported decrease in female's yield and weight as well as decrease in egg mass and eggs hatched compared to only KLH injected group. Similar results have also been reported using sea lice whole extract or lice protein involved in midgut function and blood digestion, as a vaccine in Atlantic salmon, resulting in fewer oviparous female lice and lower fecundity [23, 29]. Based on our results, it is expected that a reduction in parasite fecundity due to vaccination will have an exponential reduction effect on the overall lice population and thus the lice load on the host at later generations, and consequently will warrant a reduction in chemical or drug free treatments to control lice.

Analogous to the overall efficacy calculated for the pP0 antigen against *R. B. microplus* ticks as well as other authors [30–32], we applied a formula to our experiment for estimating vaccine efficacy in order to evaluate the impact of the vaccine candidate on the lice fecundity as well as on the hatching and survival of F1 copepodids. Based on this formula, vaccinated group 2 has obtained an overall vaccine efficacy of 56% whereas group 3 showed 25%, suggesting the utility of the vaccine candidate through ip method only. However, further in-depth work has to be done. Moreover, lice count as a proxy for resistance had been questioned, since individual lice counts vary between trials and certain immune genes are affected negatively by increasing number of lice [33]. Therefore, large number of experimental animals and experimental tanks must be used in these types of immunization and challenge trials and treatment efficacy parameters other than lice count should be considered.

In Atlantic salmon, normally IgM transcripts are most abundant followed by IgT, especially in spleen and head kidney [34]. In the present study, the increase in relative expression level of IgM and IgT in spleen, head kidney and skin in vaccinated groups, indicated their important role in systemic and mucosal immune response in the context of copepodids infestation. In agreement with these results, Tadiso et al. observed 10-fold increases in IgT expression in the skin from infested Atlantic salmon and up-regulation of IgT and IgM in spleen and skin two weeks post lice infection [34], but until now, IgM and IgT responses observed in Atlantic salmon have not been associated with protection against copepodids infestation. The role of antibodies in protection against copepodids infestation in teleost has not been fully explored and needs further understanding. For future studies, it will be of greater importance to measure antigen-specific IgM in serum and in mucus by ELISA, to understand their role in protection and crosstalk during salmon lice infestation, post vaccination.

To understand the underlying immune mechanism, we assessed transcriptomic responses at systemic and local level in immunized salmon focusing on mid and late response post infestation. The results showed substantial increase in relative expression of pro-inflammatory mediators (IL-1β, TNF-α, IL-8) at the systemic level (spleen and to some extent in head kidney). This is in line with the sustained response of systemic pro-inflammatory cytokines seen in the more resistant species such as the pink salmon throughout the infection and even after rejection in these fish [35]. Barker et al. (2019) also obtained similar results with significantly

higher levels of IL-1β expression at 17 dpi with sea lice [36]. The same pattern of expression held true when investigating tissue repair enzyme MMP 9 gene expression that was used as an indicator to evaluate the wound healing response of the fish to sea lice infestation. The increase in MMP 9, had been suggested by several groups as a possible mechanism for sea lice resistance in Atlantic salmon [33, 36, 37]. In addition, induced high IL-8 transcript levels in skin and spleen post vaccination (69 d) has been implicated as an inducer of neutrophil migration and antibody secreting cells locally. Furthermore, it can also be speculated that elevated systemic expression of inflammatory and T regulatory mediators, pre and post lice challenge in the vaccinated fish compared to only adjuvant control, might have been involved in local expression of IgM and IgT transcript. Moreover, early upregulation of immunoglobulin like genes in spleen, head kidney and skin, in addition to panels of immune genes, indicates a rapid activation of the systemic as well as local anti-parasitic response to some extent, which is in accordance with the results obtained by Skugor et al. (2008) [38]. This demonstrates a facilitated cross talk between immune genes in vaccinated group pre and post infection.

On the other hand, the pro-inflammatory response in skin post infestation appeared to be at the basal level compared to adjuvant control, except for IL-8, which was significantly upregulated at 17 dpi and 28 dpi of sampling for both vaccination types. It is possible that by the time systemic inflammatory response was mounted, the cytokine expression had already returned to its basal level in skin. A microarray experiment looking at the effects of early stage *L. salmonis* attachment showed that the local expression in skin decreased at early time points from 5 dpi, although the systemic response in the spleen remained throughout the study period [34]. As the earliest samples for gene expression in our study was taken at 17 dpi, it is possible that early transient increase of inflammatory cytokines in the skin was missed. Another possible explanation can be the sampling of skin from the standard area of the fish (near the dorsal fin and above the lateral line), regardless of louse attachment. Therefore, if the cutaneous inflammatory response is directed exclusively at the site of attachment, it would not have been targeted by the standardized skin sampling, especially if infection intensity was not as dramatic as those reported previously [34]. Matrix metalloprotease plays a role in the reconstruction process of the extracellular matrix during wound healing. In sea lice infested salmon, the slow repair of extracellular matrix is in parallel with stable up-regulation of MMP9 and MMP13 at the damaged sites, and whose excessive activity may contribute to the development of chronic wounds [38]. Here, absence of MMP-9 stimulation in skin could suggest less damage to the host with no chronic wound and subsequently less tissue repair required. This was confirmed by no visible damages to the skin during the experiment.

Despite that, immersion bath stimulates immune response, mainly in mucosal tissues such as skin [39]. The intraperitoneal injection of the candidate vaccine plus immersion bath with inclusion bodies received by group 3 was less effective, although some immune parameters were improved. The response to other parasites has often been described in terms of Th1/Th2 dichotomy, but recent studies have shown that host-pathogen interactions are more complex. A T cell effector subset Th17, characterized by the production of IL-17 and IL-22, were identified along with signature cytokines for regulatory T cell subset (T reg), being inhibitory IL-10 and/or TGF-β. Th1, Th2 and Th17 reciprocally regulates the development and function of each other, while T reg cells suppress all three subsets [34, 38, 40]. The regulatory cytokines control inflammation and thus protect against immunopathology, but in doing so they reduce the effectiveness of immune mechanisms responsible for the expulsion of the parasites. Here, pro-inflammatory response in skin might be regulated by IL-10, IL4 and IL-22 at 28 dpi of the immunized salmon but requires further study to match with the earlier time points than 17dpi. This is somewhat in accordance with the results obtained in resistant coho salmon (*Oncorhynchus kisutch*), although at an earlier time-point up to 72 hours [41]. We observed

down regulation of IL-22 and IL-10 in spleen of salmon at chalimus stage of infection and an increase in IL-1β, TNF-α and IL-8 at subsequent pre-adult stage in group 2. In group 3, down regulation of IL-22 and IL-10 was seen in spleen at pre-adult stage of infection (28 dpi), which in turn is related to the increase observed in pro-inflammatory cytokines at adult stage (50 dpi). These differences in the regulation of inflammation could explain the differences found in the results between different vaccination methods i.e group 2 and group 3. Further studies targeting more immunological markers could clarify the mechanisms responsible for the differences between the two groups.

Previous studies have shown that the pathological effects of sea lice become especially profound for the host fish when they reach mobile stage on the host compared to the attached chalimus stage [42]. This explores the important strategy the host should develop to avoid damage on the skin through early free-ranging pre-adult lice interaction and develop resistance against it. The use of hierarchical clustering heat map and PCA analysis in this study showed a clear overview of the gene expression in different tissues across the groups at different time-points post infection and the way the genes were regulated in the vaccinated and the non-vaccinated group. Most of the genes were highly to moderately upregulated at 28 dpi in only ip vaccinated, group 2, when the infestation was at the mobile stage (pre-adult), while they were upregulated to some extent in the ip plus bath vaccinated group 3 at both chalimus and pre-adult phase. In addition, in only ip vaccinated group 2, differential gene expression, cluster analysis and principal component analysis also showed the dynamics of T-cell response as mixed Th1/Th2/T17/Treg at the pre-adult lice stage of infestation. This reflects the importance and potential of the gene modulation strategy of only ip vaccinated group compared to ip plus bath vaccinated group, against the early mobile lice stage, for effective vaccine efficacy at the later adult stage and that correlated well with the adult female lice count and fecundity data documented in the ip injected vaccine group.

Taken together, our result provided new insight into the potential of the candidate vaccine in reducing salmon lice load and its effect on host-parasite interaction with minimal side-effects. The calculated vaccine efficacy of 56% in the ip injected vaccine group suggests a larger impact on F1 parasite generation by reduced re-infection loads via fewer females and decreased fecundity. In addition, the results revealed the priming of immune response post vaccination and pre-challenge, leading to simultaneous involvement of both systemic and local immunity during the salmon lice interaction for vaccinated fish, at the mobile lice stages. These findings provided valuable leads for the effectiveness of the P0 antigen as a vaccine candidate against salmon lice (*L. salmonis*). However, long-term challenge trials with higher number of fish per tank and studies of re-infection post vaccination is necessary to fully understand and explore the protection potential of a candidate vaccine and underlying molecular mechanism of protection at the gene level. Another aspect to consider is that in experimental challenge conditions, the infestation load is usually very high (i.e in this validation study: 35 copepodids per fish) and is far higher compared to the natural conditions in the field. Therefore, performing a challenge experiment considering these points will be the next step for additional evaluation of the vaccine efficacy in controlling salmon lice infestation.

## Supporting information

**S1 Fig. Two-way hierarchical clustering heat map for each tissue.** The rows represent gene expression and the column represents different sampling points within respective groups. Group details: Group 1 is control group; Group2 received ip injection of adjuvant emulsified vaccine antigen; Group 3 received ip injection of adjuvant emulsified vaccine antigen + bath immunization.
(PDF)

**S2 Fig. PCA analysis of head kidney samples post immunization and lice infestation.** PCA analysis of head kidney samples from vaccinated (group 2 and 3) and only adjuvant vaccinated (group 1) groups at 0, 17, 28 and 50 dpi (A and B). Analysis was based on mean fold-changes of all genes for each individual sample at each sampling point (smaller symbols) relative to the unvaccinated control. The ellipses indicate the group dispersion/variability from the centroid (larger symbols) calculated using all individual fold-changes values/group (A). (C) shows the contribution of sampling points to different components. (D) shows the contribution of genes on different components and the significant genes contributing in principal component 1 and 2. (PDF)

**S3 Fig. PCA analysis of spleen samples post immunization and lice infestation.** PCA analysis of spleen samples from vaccinated (group 2 and 3) and only adjuvant vaccinated (group 1) groups at 0, 17, 28 and 50 dpi (A and B). Analysis was based on mean fold-changes of all genes for each individual sample at each sampling point (smaller symbols) relative to the unvaccinated control. The ellipses indicate the group dispersion/variability from the centroid (larger symbols) calculated using all individual fold-changes values/group (A). (C) shows the contribution of sampling points to different components. (D) shows the contribution of genes on different components and the significant genes contributing in principal component 1 and 2. (PDF)

**S4 Fig. Transcriptional analysis of immune genes post immunization and lice infection.** Transcript levels of the pro-inflammatory cytokines (A) and immune genes (B) in spleen, head kidney and skin at different sampling points: 69 days from first vaccination (69d) or zero day challenge and after challenge (dpi: days post infestation), were analysed by real-time QPCR. The QPCR data were normalized to the geometric mean of the 3 house-keeping genes (EF-1a, 18S and β-actin) and expression is relative to the pre-immunized level. Fold change was calculated using the primer efficiency. Data shown represent the mean ± SD of experiments performed in triplicate, n = 18 fish/group (6 fish/replicate). Statistical analysis was carried out using one-way ANOVA or Kruskal Wallis test followed by Tukey or Dunn's Multiple Comparison compared to control group ($^*P < 0.05$, $^{**}P < 0.01$, $^{***}P < 0.001$). Group details: Group 1 is control group; Group2 received ip injection of the adjuvant emulsified vaccine antigen; Group 3 received ip injection of adjuvant emulsified vaccine antigen + bath immunization. (PDF)

**S1 Table. Observation noted on day 8 post incubation of egg strings for assessing the hatching efficiency and visual health status of the hatched copepodids.** Fifty egg strings (sampled from the first reproductive event at 50 dpi) from each experimental group were randomly distributed and incubated in 5 parallel aerated flow-through incubators (containing 500 mL filtered seawater/incubator at ~10°C) having 10 egg strings in each incubator. Group details: Group 1 is control group; Group 2 received ip injection of adjuvant emulsified vaccine antigen; Group 3 received ip injection of adjuvant emulsified vaccine antigen + bath immunization. (TIF)

**S1 Dataset.** Sheet 1: Lice count and fecundity raw data. Sheet 2: Mean data used for global gene expression analysis in different tissues: Skin, Spleen, and Head kidney. (XLSX)

## Acknowledgments

We would like to thank staffs at Aquaculture Research Station in Tromsø for assistance in fish maintenance, copepodid production, performing lice challenge and lice counting. We also

thank Dr. Trilochan Swain for providing valuable suggestions during the development of project concept and manuscript preparation.

## Author Contributions

**Conceptualization:** Jaya Kumari Swain, Yamila Carpio, Mario Pablo Estrada.

**Data curation:** Jaya Kumari Swain, Ajey Kumar.

**Formal analysis:** Jaya Kumari Swain, Yamila Carpio, Ajey Kumar.

**Funding acquisition:** Jaya Kumari Swain.

**Investigation:** Jaya Kumari Swain, Yamila Carpio, Lill-Heidi Johansen, Janet Velazquez, Liz Hernandez, Yeny Leal.

**Methodology:** Jaya Kumari Swain, Yamila Carpio.

**Project administration:** Jaya Kumari Swain.

**Resources:** Jaya Kumari Swain, Yamila Carpio, Mario Pablo Estrada.

**Supervision:** Jaya Kumari Swain, Yamila Carpio.

**Validation:** Jaya Kumari Swain, Yamila Carpio.

**Visualization:** Jaya Kumari Swain, Yamila Carpio.

**Writing – original draft:** Jaya Kumari Swain, Yamila Carpio.

**Writing – review & editing:** Jaya Kumari Swain, Yamila Carpio, Lill-Heidi Johansen, Ajey Kumar, Mario Pablo Estrada.

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
