## [Decision Letter · Decision Letter 0]

22 Jul 2020

PONE-D-20-16484

Impact of a candidate vaccine on the dynamics of salmon lice (Lepeophtheirus salmonis) infestation and immune response in Atlantic salmon (Salmo salar L.)

PLOS ONE

Dear Dr. Swain,

Thank you for submitting your manuscript to PLOS ONE. After careful consideration, we feel that it has merit but does not fully meet PLOS ONE’s publication criteria as it currently stands. Therefore, we invite you to submit a revised version of the manuscript that addresses the points raised during the review process.

It was reviewed by two experts in the field who have suggested some revisions be made prior to acceptance.

If you could write a response to reviewers that would expedite revision upon resubmission.

I wish you the best of luck with your revisions.

Hope you are keeping safe and well in these difficult times.

We look forward to receiving your revised manuscript.

Kind regards,

Simon Clegg, PhD

Academic Editor

PLOS ONE

We note that one or more of the authors are employed by a commercial company: Nofima AS.

2.1. Please provide an amended Funding Statement declaring this commercial affiliation, as well as a statement regarding the Role of Funders in your study. If the funding organization did not play a role in the study design, data collection and analysis, decision to publish, or preparation of the manuscript and only provided financial support in the form of authors' salaries and/or research materials, please review your statements relating to the author contributions, and ensure you have specifically and accurately indicated the role(s) that these authors had in your study. You can update author roles in the Author Contributions section of the online submission form.

2.2. Please also provide an updated Competing Interests Statement declaring this commercial affiliation along with any other relevant declarations relating to employment, consultancy, patents, products in development, or marketed products, etc. 

Reviewers' comments:

Reviewer's Responses to Questions

**Comments to the Author**

1. Is the manuscript technically sound, and do the data support the conclusions?

Reviewer #1: Yes

Reviewer #2: Partly

2. Has the statistical analysis been performed appropriately and rigorously? 

Reviewer #1: Yes

Reviewer #2: No

3. Have the authors made all data underlying the findings in their manuscript fully available?

Reviewer #1: Yes

Reviewer #2: No

4. Is the manuscript presented in an intelligible fashion and written in standard English?

Reviewer #1: Yes

Reviewer #2: Yes

5. Review Comments to the Author

Reviewer #1: PONE-D-20-16484

Impact of a candidate vaccine on the dynamics of salmon lice (Lepeophtheirus salmonis) infestation and immune response in Atlantic salmon (Salmo salar L.)

General comments:

The manuscript is well written. It describes the research performed and results interpreted of the immune response of vaccinated Atlantic salmon to two sea lice vaccine groups as compared to a control quite thoroughly. Developing an efficacious vaccine to control sea lice infestations in Atlantic salmon would greatly enhance the sustainability of the Atlantic salmon farming industry globally and describing and defining the immune response as it relates to antibody and gene regulation is critical to advances in vaccine development. However, I don’t feel concluding that the results support the effectiveness of a vaccine candidate is accurate if effectiveness is defined as ‘the degree to which something is successful in producing a desired result’. The manuscript is valuable for the information presented with the specific type of vaccine candidates and vaccination strategy but conclusions should be more reflective of actual protection against sea lice infestation.

Specific comments:

Abstract:

Lines 36 & 37: Do the overall results support effectiveness? For consideration, the following definitions are provided.

Efficacy can be defined as the performance of an intervention under ideal and controlled circumstances, whereas effectiveness refers to its performance under 'real-world' conditions.

Introduction:

Line 59: suggest replacing might with can

Line 116: correct font and delete one period

Fish immunization and lice challenge

It might flow better to change this section to 3 sections, Fish husbandry, Fish immunization, and Lice challenge. Also, a figure that outlines fish numbers groups and fish to tank movements would make things much clearer.

Days post immunization could be described using the number of degree days post immunization???

Line 129: delete period after (24)

Sampling and lice counting

Line 159: was the dose of benzocaine a lethal dose?

Line 163: maybe use ‘humanely euthanized’ in place of killed

Gene expression studies

Line 204: Should this be a larger font as it appears to be new section? Or put in with the side effects analysis

Results

Impact of vaccine candidate post lice infestation

Lines 263 through 266: I almost think the effect of lice infestation on the fish should be its own brief section

Line 286: I still feel the use of the term efficacy isn’t appropriate

Global assessment: Heat map and hierarchical clustering

Line 315: map should be maps

Line 320: change infection to infestation

Principle component analysis (PCA)

Line 344: change infection to infestation

Discussion

Line 390: change infection to infestation

Line 393: suggestion only- change ‘is still on its way’ to ‘forthcoming’

Line 394: either ‘a vaccine candidate’ or ‘vaccine candidate’

Line 394: remove comma after lead

Lines 417-418: the tense of the sentence needs to be consistent, change was to is in line 417

Lines 421 -432: I do question the interpretation again of efficacy??? I do agree larger number of experimental fish are needed for evaluating efficacy.

Line 425: need a period after only

Line 442: delete duplicate ‘in’

Line 468: should it read ‘A microarray experiment’ or “Microarray experiments’’

Line 486: what are you suggesting in this line, did the immune priming make the response worse or did it overstimulate and exhaust the response??

Line 534: suggestion only- change ‘have in mind’ to ‘consider’

Line 537: I’m not convinced the study is ready for a field study but rather repeated with larger number of fish

Reviewer #2: Mostly well done, and certainly a valuable contribution on an important topic. There are some major and minor concerns that must be addressed, but afterwards, I think this manuscript would be suitable for publication.

Most importantly, I worry that the authors have used an inappropriate formula in calculating the overall vaccine effect, which leads to a misleading finding (see comment on L186). The numbers in Table 2 are fine, but in the text, the authors report 86% overall efficacy in Group 2, yet according to Table 2 there is only a 40% reduction in the number of adult females relative to the control group, and only a 23% reduction in the reproductive success of those adult females (i.e. F1 larval counts). To me this means a 54-55% reduction overall depending on whether you consider all adult females or only those with eggs.

The writing is mostly easy to understand, but the quality could be further improved by some grammatical corrections throughout.

Abstract

Good, although I would like to see an overall effect size given in the abstract. “Good potential” could probably mean anything from 10% to 90% reduction!

Introduction

A good background, well done.

Methods

Antigen purification: I am not an expert in these methods so cannot really comment on their suitability, but I see no obvious errors.

L126-127: Give a full overview of the experimental design upfront here. I also don’t see what size the experimental tanks were – 500 L seems small for 120 fish? I suggest something like “Six XXX L tanks were stocked with 120 fish each, with XXX tanks assigned to each of the 3 experimental groups: procedural control (Group 1), injected vaccine (Group 2), and injected vaccine + bath immunization (Group 3).” This way it is easier for the reader to understand the basic design and sample size at the fish and tank level. Details about the treatments and rearing conditions can come afterwards.

L148: Were oxygen levels monitored during this time? Depending on the size of the tanks, oxygen levels could get very low during an hour without flow.

L156: Earlier it is said that experimental groups were kept in duplicate. I assumed that to mean 2 tanks per group. Here it is 10 fish per tank = 30 fish per group, which is repeated further on in the paper.

L162: It is good that counting was done this way, but how were fish sampled? My experience is that most lice are actually lost during netting and anaesthesia rather than during counting. This paragraph should say how the sampling took place. For example, were the tanks sedated before sampling (this reduces the number of lice lost during netting), were fish netted or removed some other way (again, affects how many lice are lost), were the 10 fish per tank anaesthetised together in a bucket or separately, was the anaesthetic water checked for detached lice, etc.). This shouldn’t qualitatively change the vaccine effect, but it may change our understanding of infestation density across all three groups (especially because lice levels decreased markedly over time).

L172: How were the egg strings from different groups distributed across the 5 incubators? Can you be sure that differences between groups were not caused by differences between incubators?

L186: This formula seems problematic to me: “Vaccine efficacy (%) = 100 x [1-(NCh x NPA x NF x NM x FE x NE x LE x CC)]”. Unless I misunderstand, it looks like repeated counting of the same effect. For example, there is no need to include the immature stages here – losses of lice at all stages will accumulate and be ultimately represented by the difference in the density of adult females. You can then multiply that effect by effects on reproductive output per female. The study cited (Rodríguez-Mallon et al. 2015) used a similar formula but using different parameters, so does not support the usage here. I also disagree with the use of egg string number (NE), as this is more likely to be a sampling artefact than a biological effect (see also L256). Moreover, wouldn’t F1 copepodid counts also incorporate differences in egg string length (LE) if that variable is important? Unless the authors can explain why this isn’t repeated counting of the same effects, I recommend using a much simpler and more defensible formula to calculate an overall effect, representing how many female lice survive to maturity and how many larvae they produce:

Vaccine efficacy (%) = 100 x [1-(FE x CC)]. Note, the CC parameter should be corrected for the number of egg strings collected per female (copepod count / number of egg strings).

L202: K is usually based on a simple formula, can it be included here to save the reader looking up Barnham and Baxter to find out how K was calculated?

L220: “Experimental groups were conducted in triplicates”. I’m not clear on what this means for the statistical models. I would like to see a bit more explanation of how the models were specified. I assume a different model for each stage? Were there any factors other than the experimental group? Were individual fish, or tanks, treated as the unit of replication? If fish are replicates, did you test tank identity as a factor? If tanks are replicates, how were lice densities on individual fish summarised to tank level?

L221: Outliers in which variable (lice levels?), and why? What was the criteria for identifying an outlier and is it biologically justified? How many were removed? Does leaving them in change the outcome?

Results

L234-236: Doesn’t this belong in the methods rather than results?

L236: Thank you for providing SDs (not always done in aquaculture!). Can you clarify in the text at what level they were calculated? e.g. SD across all fish within a group, or across tank means within a group? Some more details on the statistical results would also improve the quality (e.g. degrees of freedom, test statistics), or else include the model outputs in the supplementary material.

L256: I might have missed this, but what was done when lice had only one egg string vs two egg strings? Was this accounted for when comparing copepodid densities? Egg strings are very easily dislodged during sampling so I would correct for missing egg strings rather than considering it to be a result of the treatment. To me, there is no biological basis for the comparison of Panels E, F and G in Figure 2. It more likely to be comparing how roughly fish were handled than any effect of a vaccine.

L291: Give % differences in the text here – this is an important result when considering future commercial applications.

L313 onwards: Very interesting results, well done.

Discussion

L392-395: I’m not sure this sentence adds much of substance, I would consider deleting.

L407-408: Does it make more sense to talk in terms of a cumulative effect here?

L421-432: This paragraph is really difficult to follow (both the logic and the language). I suggest rewriting it in a more methodical manner. Importantly, I don’t see how a “careful analysis of formulas” affects our understanding of how well vaccines work. Ultimately, doesn’t it come down to how many lice remain attached with how well they reproduce? I guess this stems from the issues with the formula that I discuss above.

Related points:

L422: Replace “extrapolated” with “applied”.

L430: Please add a note that a large number of tanks is also important (fish within tanks are not truly independent replicates, as they often vary together).

L483 onwards: Interesting explanation. Can you also discuss how likely is that the result is spurious (due to low levels of replication at the tank level) versus the mechanisms you outline here? For example, you could point out that relative infestation densities changed over time, rather than being consistently different from day 1. Perhaps somewhere you could give the mean infestation density for every tank at the earliest possible day (I guess that would be 17 dpi?) so the reader can assess how tanks varied within groups (in more detail than SD provides).

L508: I would simplify to “…reach the mobile stage on the host…”. “Mobile” is the term used almost universally used in the literature.

Figures

Generally okay.

See comment above regarding panels E-G in Figure 2.

If you want to save space in the article, the table of observations within Figure 3 could be moved to the supplement.

Data

I don’t see a link for data sharing – apologies if I missed it.

6. PLOS authors have the option to publish the peer review history of their article (what does this mean?). If published, this will include your full peer review and any attached files.

Reviewer #1: No

Reviewer #2: No

---

## [Author Response · Author response to Decision Letter 0]

13 Aug 2020

Response to Academic Editor 

Author’s response: We have gone through the templates and revised the manuscript accordingly. 

We note that one or more of the authors are employed by a commercial company: Nofima AS.

Author’s response: Nofima is not a commercial company. Nofima is a food research institute and is a non-profit research institution. We have added this statement in the Competing interests section.

Updated Competing Interests: “Nofima is a non-profit research institution. The authors have declared that no competing interests exist”.

2.1. Please provide an amended Funding Statement declaring this commercial affiliation, as well as a statement regarding the Role of Funders in your study. If the funding organization did not play a role in the study design, data collection and analysis, decision to publish, or preparation of the manuscript and only provided financial support in the form of authors' salaries and/or research materials, please review your statements relating to the author contributions, and ensure you have specifically and accurately indicated the role(s) that these authors had in your study. You can update author roles in the Author Contributions section of the online submission form.

Author’s response: The updated funding Statement includes: “Funding for this work was provided to JKS by Norwegian Seafood Research Fund (Fiskeri- og Havbruksnæringens Forskningsfond, FHF), grant number 901461. The funder had no role in the study design, data collection and analysis, decision to publish, or preparation of the manuscript”.

2.2. Please also provide an updated Competing Interests Statement declaring this commercial affiliation along with any other relevant declarations relating to employment, consultancy, patents, products in development, or marketed products, etc. 

Author’s response: Since Nofima is a non-profit research institution, the above mentioned declaration is not applied.

Author’s response: The updated Funding Statement and Competing Interests Statement are mentioned in response 2 and 2.1 and in the cover letter.

Response to the Reviewers

Review Comments to the Author

Reviewer #1: PONE-D-20-16484

Impact of a candidate vaccine on the dynamics of salmon lice (Lepeophtheirus salmonis) infestation and immune response in Atlantic salmon (Salmo salar L.)

General comments:

The manuscript is well written. It describes the research performed and results interpreted of the immune response of vaccinated Atlantic salmon to two sea lice vaccine groups as compared to a control quite thoroughly. Developing an efficacious vaccine to control sea lice infestations in Atlantic salmon would greatly enhance the sustainability of the Atlantic salmon farming industry globally and describing and defining the immune response as it relates to antibody and gene regulation is critical to advances in vaccine development. However, I don’t feel concluding that the results support the effectiveness of a vaccine candidate is accurate if effectiveness is defined as ‘the degree to which something is successful in producing a desired result’. The manuscript is valuable for the information presented with the specific type of vaccine candidates and vaccination strategy but conclusions should be more reflective of actual protection against sea lice infestation.

Specific comments:

Abstract:

Lines 36 & 37: Do the overall results support effectiveness? For consideration, the following definitions are provided.

Efficacy can be defined as the performance of an intervention under ideal and controlled circumstances, whereas effectiveness refers to its performance under 'real-world' conditions.

Authors’ response: The reviewer is right. We changed the word «effectiveness» for «efficacy» which is more appropriate for a lab scale experiment.

Introduction:

Line 59: suggest replacing might with can. 

Line 116: correct font and delete one period.

Authors’ response: The reviewer suggestion have been considered.

Fish immunization and lice challenge

It might flow better to change this section to 3 sections, Fish husbandry, Fish immunization, and Lice challenge. 

Authors’ response: It was changed according reviewer suggestion.

Also, a figure that outlines fish numbers groups and fish to tank movements would make things much clearer.Days post immunization could be described using the number of degree days post immunization??? 

Authors’ response: All these suggestions were taken into account to modify Fig 1.

Line 129: delete period after (24) 

Authors’ response: We have modified the bracket as this was the reference number.

Sampling and lice counting

Line 159: was the dose of benzocaine a lethal dose? 

Authors’ response: Yes, fish was given a lethal dose of anaesthesia (0.01% benzocaine) before counting. 

Line 163: maybe use ‘humanely euthanized’ in place of killed. 

Authors’ response: Modified as suggested.

Gene expression studies

Line 204: Should this be a larger font as it appears to be new section? Or put in with the side effects analysis 

Authors’ response: Thanks for mentioning it. Yes, gene expression studies should be a larger font. This have been corrected.

Results

Impact of vaccine candidate post lice infestation

Lines 263 through 266: I almost think the effect of lice infestation on the fish should be its own brief section 

Authors’ response: We have mentioned those lines in a separate paragraph. 

Line 286: I still feel the use of the term efficacy isn’t appropriate

Authors’ response: Taking into account the definition given by the reviewer above that «Efficacy» can be defined as the performance of an intervention under ideal and controlled circumstances, I think is appropriate.

Besides, terminology “vaccine efficacy” used here should not be interpreted as protection obtained. The term used was based on the revised formula that was aimed to evaluate the impact of vaccination on female lice fecundity. The formula of the vaccine efficacy was revised and added in the revised text in Line 204.

Global assessment: Heat map and hierarchical clustering

Line 315: map should be maps: Line 320: change infection to infestation: 

Authors’ response: This is fixed

Principle component analysis (PCA)

Line 344: change infection to infestation : 

Authors’ response: This is changed.

Discussion

Line 390: change infection to infestation 

Line 393: suggestion only- change ‘is still on its way’ to ‘forthcoming’ 

Authors’ response: The word ‘Infection’ is changed to ‘infestation’. Thanks for the suggestion for line 393. We have deleted this sentence as per reviewer 2 suggestion. 

Line 394: either ‘a vaccine candidate’ or ‘vaccine candidate’ 

Line 394: remove comma after lead

Lines 417-418: the tense of the sentence needs to be consistent, change was to is in line 417: Lines 421 -432: I do question the interpretation again of efficacy??? I do agree larger number of experimental fish are needed for evaluating efficacy. 

Authors’ response: Lines 394 and 417 were revised as suggested. Related to line 421-432, we agree with the reviewer. We have modified this paragraph. Following changes were made in the revised text in L460 - 466: 

“Analogous to the overall efficacy calculated for the pP0 antigen against R. B. microplus ticks as well as other authors [31-33], we applied a formula to our experiment for estimating vaccine efficacy in order to evaluate the impact of the vaccine candidate on the lice fecundity. Based on this formula, vaccinated group 2 has obtained an overall vaccine efficacy of 58 % whereas group 3 showed 20 %, suggesting the utility of the vaccine candidate through ip method only “. 

Regarding the number of experiment fish, we have this statement in the text: “Therefore, large number of experimental animals and experimental tanks must be used in immunization and challenge trials”

Line 425: need a period after only 

Line 442: delete duplicate ‘in’ 

Line 468: should it read ‘A microarray experiment’ or “Microarray experiments’’ 

Authors’ response: Line 425 and 442 has been fixed as suggested. “A microarray experiment”- It has been fixed.

Line 486: what are you suggesting in this line, did the immune priming make the response worse or did it overstimulate and exhaust the response?? 

Authors’ response: The sentence “This could be the effect of exacerbated immune priming” was removed to avoid confusion. 

We think that immune priming with immersion bath made the response worse and delayed. 

We observed down regulation of IL-22 and IL-10 in spleen of salmon at chalimus stage of infection and an increase in IL-1β, TNF-α and IL-8 at subsequent pre-adult stage in group 2. In group 3, down regulation of IL-22 and IL-10 was seen in spleen at pre-adult stage of infection (28 dpi), which in turn is related to the increase observed in pro-inflammatory cytokines at adult stage (50 dpi). The differential cytokine regulation could explain the differences found in the results between different vaccination methods i.e group 2 and group 3. Further studies targeting more immunological markers could clarify the mechanisms responsible for the differences between the two groups. 

Line 534: suggestion only- change ‘have in mind’ to ‘consider’

Line 537: I’m not convinced the study is ready for a field study but rather repeated with larger number of fish 

Authors’ response: We took these into consideration and modified the sentence accordingly in L580 – 582 of the revised manuscript.

“Therefore, performing a challenge experiment considering these points will be the next step for additional evaluation of the vaccine efficacy in controlling salmon lice infestation”.

Reviewer #2: Mostly well done, and certainly a valuable contribution on an important topic. There are some major and minor concerns that must be addressed, but afterwards, I think this manuscript would be suitable for publication.

Most importantly, I worry that the authors have used an inappropriate formula in calculating the overall vaccine effect, which leads to a misleading finding (see comment on L186). The numbers in Table 2 are fine, but in the text, the authors report 86% overall efficacy in Group 2, yet according to Table 2 there is only a 40% reduction in the number of adult females relative to the control group, and only a 23% reduction in the reproductive success of those adult females (i.e. F1 larval counts). To me this means a 54-55% reduction overall depending on whether you consider all adult females or only those with eggs.

The writing is mostly easy to understand, but the quality could be further improved by some grammatical corrections throughout.

Abstract

Good, although I would like to see an overall effect size given in the abstract. “Good potential” could probably mean anything from 10% to 90% reduction! 

Authors’ response: The effect size has been included in the abstract.

Introduction

A good background, well done.

Authors’ response: We appreciate the positive comment by the Reviewer.

Methods

Antigen purification: I am not an expert in these methods so cannot really comment on their suitability, but I see no obvious errors.

L126-127: Give a full overview of the experimental design upfront here. I also don’t see what size the experimental tanks were – 500 L seems small for 120 fish? I suggest something like “Six XXX L tanks were stocked with 120 fish each, with XXX tanks assigned to each of the 3 experimental groups: procedural control (Group 1), injected vaccine (Group 2), and injected vaccine + bath immunization (Group 3).” This way it is easier for the reader to understand the basic design and sample size at the fish and tank level. Details about the treatments and rearing conditions can come afterwards. 

Authors’ response: The sentences was modified and added as suggested in the revised version of the manuscript. Tank sizes were also included wherever necessary.

L132-135: “Three 500 L tanks were stocked with 120 fish each, one tank assigned to each of the 3 experimental groups: procedural control (Group 1), injected vaccine (Group 2), and injected vaccine + bath immunization (Group 3). Each tank was supplied with continuous circulating water flow throughout the experimental period and oxygen level and temperature were recorded daily”.

The fish was stocked at a recommended density of 10 kg/m3 and this density was maintained throughout the experiment, irrespective of the tank size.

L148: Were oxygen levels monitored during this time? Depending on the size of the tanks, oxygen levels could get very low during an hour without flow.

Authors’ response: The oxygen level was monitored to keep the level stable during the lice challenge when the water flow was stopped for an hour. Aeration of water was done when required to maintain the oxygen level between 80 – 90 %.

L156: Earlier it is said that experimental groups were kept in duplicate. I assumed that to mean 2 tanks per group. Here it is 10 fish per tank = 30 fish per group, which is repeated further on in the paper. 

Authors’ response: Tank distribution has been included in Figure 1 for better understanding. Post challenge, each experimental group was split in three replicates. Thus, this is the reason that sampling of 10 fish per group is 30 fish. 

Moreover, with time, the size and weight of the fish was increasing in sea-water and post challenge experiment lasted for another 50 days. Therefore, to maintain the fish density of 10 kg/m3, we split each group in three replicate tanks of 500L capacity. Throughout the experiment, the stocking density was maintained around 10 kg/m3, irrespective of the tank size.

L162: It is good that counting was done this way, but how were fish sampled? My experience is that most lice are actually lost during netting and anaesthesia rather than during counting. This paragraph should say how the sampling took place. For example, were the tanks sedated before sampling (this reduces the number of lice lost during netting), were fish netted or removed some other way (again, affects how many lice are lost), were the 10 fish per tank anaesthetised together in a bucket or separately, was the anaesthetic water checked for detached lice, etc.). This shouldn’t qualitatively change the vaccine effect, but it may change our understanding of infestation density across all three groups (especially because lice levels decreased markedly over time). 

Authors’ response: As per reviewer suggestion, the paragraph was modified accordingly in L170-178 as mentioned below:

 “Fish were taken out one at a time gently by handle fishing net from the stocking tanks and transferred in a bucket containing an overdose of anaesthetic water (0.01% benzocaine). Care was taken for the minimum loss of lice through netting. The net was simultaneously checked for the detached lice. One fish per bucket was anaesthetized before lice counting. To avoid counting error of detached lice due to anaesthesia and handling, counting of chalimus at 17 dpi, pre-adults at 28 dpi and adults at 50 dpi on individual parasitized fish were performed under water in a white tray. After lice counting from each fish/tray, the remaining water in the respective tray and bucket were checked for detached lice.”

Moreover, we have experienced that after sedation of fish, lice is detached very quickly from the fish. Therefore, fish were taken out first one by one through handle net and then anaesthetised in a white bucket and simultaneously the net was checked for any detached lice on the net in anaesthetic water.

L172: How were the egg strings from different groups distributed across the 5 incubators? Can you be sure that differences between groups were not caused by differences between incubators?

Authors’ response: After counting the egg strings from each female lice, individual egg string was detached from the lice and its length was measured. Then the egg strings were collected in a petri dish for each group. The total no. of egg strings collected for each group was: Gr. 1 = 137, Gr. 2 = 80 and Gr. 3 = 115.

From the total no. of collected egg strings for each group as mentioned above, 50 egg strings per group were randomly distributed in 5 incubators at a rate of 10 egg strings per incubator for the hatching experiment. Each incubator contained aerated flow-through 500 mL filtered seawater at ~10 °C. The left over egg strings after the distribution in the incubator was shown in Fig 2I (revised Fig 2 F)

Since the same number of egg strings (10 nos.) were incubated in each incubator per group, differences in groups were representative of differences in hatching efficiency and fecundity.

The figure below shows the deployment of the egg strings from the petri dish into the incubator for the hatching experiment.

L186: This formula seems problematic to me: “Vaccine efficacy (%) = 100 x [1-(NCh x NPA x NF x NM x FE x NE x LE x CC)]”. Unless I misunderstand, it looks like repeated counting of the same effect. For example, there is no need to include the immature stages here – losses of lice at all stages will accumulate and be ultimately represented by the difference in the density of adult females. You can then multiply that effect by effects on reproductive output per female. The study cited (Rodríguez-Mallon et al. 2015) used a similar formula but using different parameters, so does not support the usage here. I also disagree with the use of egg string number (NE), as this is more likely to be a sampling artefact than a biological effect (see also L256). Moreover, wouldn’t F1 copepodid counts also incorporate differences in egg string length (LE) if that variable is important? Unless the authors can explain why this isn’t repeated counting of the same effects, I recommend using a much simpler and more defensible formula to calculate an overall effect, representing how many female lice survive to maturity and how many larvae they produce:

Vaccine efficacy (%) = 100 x [1-(FE x CC)]. 

Note, the CC parameter should be corrected for the number of egg strings collected per female (copepod count / number of egg strings).

Authors’ response: Taking into account the reviewer suggestion, we assumed the proposed formula of vaccine efficacy but include the parameter LE as shown below:

Vaccine efficacy (%) = 100 x [1-(FE x LE x CC)].

LE: length of egg strings in vaccinated group/length of egg strings in control group. The length of egg strings are considered representative of how many eggs were produced, because it is almost impossible to count the eggs one by one.

CC: F1 generation copepodids count from vaccinated group/F1 generation copepodids count in control group. It is an indicator of how many larvae are produced from a fixed amount of egg strings.

We think that the correction of CC is not valid since we placed the same number of egg strings per group (50 egg strings per group divided in 5 incubation chambers with 10 egg string in each one) in the incubation chambers for hatching.

With the new formula, vaccine efficacy was 58% for group 2 and 20% for group 3. This was corrected in the text.

L202: K is usually based on a simple formula, can it be included here to save the reader looking up Barnham and Baxter to find out how K was calculated? 

Authors’ response: The formula below and the related information has been included in the text.

K = (10NW)/L3

L220: “Experimental groups were conducted in triplicates”. I’m not clear on what this means for the statistical models. I would like to see a bit more explanation of how the models were specified. I assume a different model for each stage? Were there any factors other than the experimental group? Were individual fish, or tanks, treated as the unit of replication? If fish are replicates, did you test tank identity as a factor? If tanks are replicates, how were lice densities on individual fish summarised to tank level?

Authors’ response: This sentence “Experimental groups were conducted in triplicates” has been removed to avoid confusion.

For statistical analysis, to compare differences in lice number per fish among groups at each sampling point, Mann-Whitney test was performed due to unequal variances to compare vaccinated groups (Group 2 or 3) with control (Group 1). For length and weight, length of egg strings and for gene expression, analysis of variance (ANOVA) or Kruskal Wallis test was performed depending on the normal distribution and equal variance of the data followed by Tukey or Dunn’s Multiple Comparison post hoc tests. P-values < 0.05 were considered statistically significant. No other factors than experimental groups was considered for the above analysis. 

We sampled 10 fish per replicate tank, 3 replicate tanks, 30 fish per group. For statistical analysis, individual fish were used as unit of replication for each group, i.e for the comparison among groups, we used all data from 30 fish within the experimental group.

L221: Outliers in which variable (lice levels?), and why? What was the criteria for identifying an outlier and is it biologically justified? How many were removed? Does leaving them in change the outcome?

Authors’ response: We identified outliers only for the individual gene expression analysis (S4 Fig). “Prior to individual gene expression data analysis, outliers were calculated and identified using the ROUT method through Prism 6.01 software for Windows and were removed from the subsequent gene expression statistical analysis”. This information has been included in L252-255 of the revised manuscript. 

However, data analysis related to lice count and other fecundity parameters were not identified for outliers and no data was removed.

In statistics, an outlier is a data point that differs significantly from other observations. An outlier may be due to variability in the measurement or it may indicate experimental error. An outlier can cause problems in statistical analyses and thus it is sometimes wise to remove. Nevertheless, the number of outliers identified for gene expression data in our work were low, ranging from 0 to 2 outliers in total per sampling point.

Results

L234-236: Doesn’t this belong in the methods rather than results?

Authors’ response: Thanks for mentioning this. Line 234-236 has been removed.

L236: Thank you for providing SDs (not always done in aquaculture!). Can you clarify in the text at what level they were calculated? e.g. SD across all fish within a group, or across tank means within a group? Some more details on the statistical results would also improve the quality (e.g. degrees of freedom, test statistics), or else include the model outputs in the supplementary material.

Authors’ response: “SD was calculated across all fish within a group”. This sentence has been included in L248.

L256: I might have missed this, but what was done when lice had only one egg string vs two egg strings? Was this accounted for when comparing copepodid densities? 

Authors’ response: Each female with total egg string was recorded, discriminating in the counting females with two egg strings and females with one egg string. After counting the egg string, each egg string was detached from the female and its length was measured. After that egg strings were collected in a petri dish for each group as shown in Figure 2I (revised Figure no. 2F) to be used for further hatching experiment. 

For copepodid counting, as mentioned above, all egg strings were collected (no matter if they came from a female with one or two egg strings). Afterward, 10 egg string per incubator, 5 incubator per experimental groups were set for hatching. Please refer to the figure shown before.

Egg strings are very easily dislodged during sampling so I would correct for missing egg strings rather than considering it to be a result of the treatment. 

Authors’ response: Sampling was done carefully to avoid this. 

To me, there is no biological basis for the comparison of Panels E, F and G in Figure 2. It more likely to be comparing how roughly fish were handled than any effect of a vaccine.

Authors’ response: We understood the reviewer point and removed panels E, F and G from figure 2 and related text in the results section and in the figure caption.

L291: Give % differences in the text here – this is an important result when considering future commercial applications.

Authors’ response: A Line mentioning % differences has been included in line L327-329. “The reduction in weight in groups 2 and 3 compared to group 1 were 13-10%, respectively. The reduction in length were 6% in both the vaccinated groups compared to control group 1”.

L313 onwards: Very interesting results, well done.

Authors’ response: We appreciate the positive comment by the Reviewer.

Discussion

L392-395: I’m not sure this sentence adds much of substance, I would consider deleting.

Authors’ response: It has been deleted as recommended.

L407-408: Does it make more sense to talk in terms of a cumulative effect here? 

Authors’ response: The reviewer point of view was not clear to the authors.

L421-432: This paragraph is really difficult to follow (both the logic and the language). I suggest rewriting it in a more methodical manner. Importantly, I don’t see how a “careful analysis of formulas” affects our understanding of how well vaccines work. Ultimately, doesn’t it come down to how many lice remain attached with how well they reproduce? I guess this stems from the issues with the formula that I discuss above. 

Authors’ response: We rewrote the paragraph

Related points:

L422: Replace “extrapolated” with “applied”. 

Authors’ response: It has been replaced.

L430: Please add a note that a large number of tanks is also important (fish within tanks are not truly independent replicates, as they often vary together). 

Authors’ response: That’s true. It has been included as suggested in L 468 of the revised manuscript.

L483 onwards: Interesting explanation. Can you also discuss how likely is that the result is spurious (due to low levels of replication at the tank level) versus the mechanisms you outline here? For example, you could point out that relative infestation densities changed over time, rather than being consistently different from day 1. Perhaps somewhere you could give the mean infestation density for every tank at the earliest possible day (I guess that would be 17 dpi?) so the reader can assess how tanks varied within groups (in more detail than SD provides).

Authors’ response: Gene transcription profile was determined in 6 fish representative of each tank, 18 fish in total per experimental group, which is a large number of samples for qPCR analysis. In the analysis, the results were not divided depending on the tanks. The statistical analysis to score differences among groups were done using the 18 samples per group per each analysed gene and sampling point. Thus, in this way the variability in the response that could be due to tank effect, is homogenized among groups.

L508: I would simplify to “…reach the mobile stage on the host…”. “Mobile” is the term used almost universally used in the literature. 

Authors’ response: This has been fixed.

Figures

Generally okay.

See comment above regarding panels E-G in Figure 2.

If you want to save space in the article, the table of observations within Figure 3 could be moved to the supplement.

Authors’ response: As suggested, the table of observations has been removed from the Figure 3 and moved as a supplementary table S1 (S1 Table).

Data

I don’t see a link for data sharing – apologies if I missed it. 

Authors’ response: We added relevant data in the supporting information as S1 dataset. Now all relevant data are within the manuscript and its Supporting Information files.

---

## [Decision Letter · Decision Letter 1]

2 Sep 2020

PONE-D-20-16484R1

Impact of a candidate vaccine on the dynamics of salmon lice (Lepeophtheirus salmonis) infestation and immune response in Atlantic salmon (Salmo salar L.)

PLOS ONE

Dear Dr. Swain,

Thank you for submitting your manuscript to PLOS ONE. After careful consideration, we feel that it has merit but does not fully meet PLOS ONE’s publication criteria as it currently stands. Therefore, we invite you to submit a revised version of the manuscript that addresses the points raised during the review process.

Many thanks for submitting your manuscript to PLOS One

It was reviewed by the same two reviewers who reviewed the first manuscript and they have suggested some more very minor modifications be made prior to acceptance

If you could write a response to reviewers, that will help to expedite revision when you re-submit

I wish you the best of luck with your revisions

Hope you are keeping safe and well in these difficult times

Thanks

Simon

We look forward to receiving your revised manuscript.

Kind regards,

Simon Clegg, PhD

Academic Editor

PLOS ONE

Reviewers' comments:

Reviewer's Responses to Questions

**Comments to the Author**

1. If the authors have adequately addressed your comments raised in a previous round of review and you feel that this manuscript is now acceptable for publication, you may indicate that here to bypass the “Comments to the Author” section, enter your conflict of interest statement in the “Confidential to Editor” section, and submit your "Accept" recommendation.

Reviewer #1: All comments have been addressed

Reviewer #2: (No Response)

2. Is the manuscript technically sound, and do the data support the conclusions?

Reviewer #1: (No Response)

Reviewer #2: Partly

3. Has the statistical analysis been performed appropriately and rigorously? 

Reviewer #1: (No Response)

Reviewer #2: N/A

4. Have the authors made all data underlying the findings in their manuscript fully available?

Reviewer #1: (No Response)

Reviewer #2: Yes

5. Is the manuscript presented in an intelligible fashion and written in standard English?

Reviewer #1: (No Response)

Reviewer #2: Yes

6. Review Comments to the Author

Reviewer #1: (No Response)

Reviewer #2: The authors have generally done a good job of revising the manuscript, and I have only two substantive concerns remaining (one of which still affects the overall vaccine efficacy estimate). I’m confident that both concerns can be addressed to my satisfaction.

The writing could still benefit from some copy editing, but I’ll leave this for the authors and/or journal to manage.

Well done to the authors for the effort they’ve put into this trial.

1) Regarding lice and egg string loss during netting

Original comment:

L162: It is good that counting was done this way, but how were fish sampled? My experience is that most lice are actually lost during netting and anaesthesia rather than during counting. This paragraph should say how the sampling took place. For example, were the tanks sedated before sampling (this reduces the number of lice lost during netting), were fish netted or removed some other way (again, affects how many lice are lost), were the 10 fish per tank anaesthetised together in a bucket or separately, was the anaesthetic water checked for detached lice, etc.). This shouldn’t qualitatively change the vaccine effect, but it may change our understanding of infestation density across all three groups (especially because lice levels decreased markedly over time).

Authors’ response:

As per reviewer suggestion, the paragraph was modified accordingly in L170-178 as mentioned below: “Fish were taken out one at a time gently by handle fishing net from the stocking tanks and transferred in a bucket containing an overdose of anaesthetic water (0.01% benzocaine). Care was taken for the minimum loss of lice through netting. The net was simultaneously checked for the detached lice. One fish per bucket was anaesthetized before lice counting. To avoid counting error of detached lice due to anaesthesia and handling, counting of chalimus at 17 dpi, pre-adults at 28 dpi and adults at 50 dpi on individual parasitized fish were performed under water in a white tray. After lice counting from each fish/tray, the remaining water in the respective tray and bucket were checked for detached lice.” Moreover, we have experienced that after sedation of fish, lice is detached very quickly from the fish. Therefore, fish were taken out first one by one through handle net and then anaesthetised in a white bucket and simultaneously the net was checked for any detached lice on the net in anaesthetic water.

New comment:

This is a better description of the sampling protocol, although I’m still not convinced that the authors are able to prevent loss of any lice or egg strings during netting. As I noted in my original comment, I realise that this does not matter much for the measured vaccine efficacy because all groups received the same handling, but in a study that directly concerns infestation density and infestation persistence, likely sources of lice loss during sampling should be acknowledged. It is well known that netting salmon (and especially unsedated salmon) causes some adult lice and egg strings to be dislodged from abrasion by the mesh, and netting “gently” or “with care” doesn’t change this, especially if the fish is struggling. I would therefore like the authors to include a note either acknowledging this or explaining how they can be sure that none were lost. E.g. (i) maybe the net mesh was fine enough to capture lice and egg strings that might have been dislodged, (ii) maybe they are confident that they would see any dislodged egg strings in the tank after netting (detached lice would likely reattach to another host in the tank – something I’ve seen before), or (iii) acknowledge that some lice and egg strings could have been lost before the fish was transferred to the anaesthetic bucket.

A small point on wording: Change “handle fishing net” to your choice of “hand net”, “dip net” or “scoop net”

2) Regarding the vaccine efficacy formula

Original comment:

L186: This formula seems problematic to me: “Vaccine efficacy (%) = 100 x [1-(NCh x NPA x NF x NM x FE x NE x LE x CC)]”. Unless I misunderstand, it looks like repeated counting of the same effect. For example, there is no need to include the immature stages here – losses of lice at all stages will accumulate and be ultimately represented by the difference in the density of adult females. You can then multiply that effect by effects on reproductive output per female. The study cited (Rodríguez-Mallon et al. 2015) used a similar formula but using different parameters, so does not support the usage here. I also disagree with the use of egg string number (NE), as this is more likely to be a sampling artefact than a biological effect (see also L256). Moreover, wouldn’t F1 copepodid counts also incorporate differences in egg string length (LE) if that variable is important? Unless the authors can explain why this isn’t repeated counting of the same effects, I recommend using a much simpler and more defensible formula to calculate an overall effect, representing how many female lice survive to maturity and how many larvae they produce: Vaccine efficacy (%) = 100 x [1-(FE x CC)]. Note, the CC parameter should be corrected for the number of egg strings collected per female (copepod count / number of egg strings).

Authors’ response:

Taking into account the reviewer suggestion, we assumed the proposed formula of vaccine efficacy but include the parameter LE as shown below:

Vaccine efficacy (%) = 100 x [1-(FE x LE x CC)].

LE: length of egg strings in vaccinated group/length of egg strings in control group. The length of egg strings are considered representative of how many eggs were produced, because it is almost impossible to count the eggs one by one.

CC: F1 generation copepodids count from vaccinated group/F1 generation copepodids count in control group. It is an indicator of how many larvae are produced from a fixed amount of egg strings. We think that the correction of CC is not valid since we placed the same number of egg strings per group (50 egg strings per group divided in 5 incubation chambers with 10 egg string in each one) in the incubation chambers for hatching.

With the new formula, vaccine efficacy was 58% for group 2 and 20% for group 3. This was corrected in the text.

New comment:

I agree with the authors that CC should not be corrected for the number of egg strings (it is now clear that the same number of egg strings were used per group).

However, the inclusion of LE still seems incorrect. I agree that LE is a fair proxy for number of eggs, and is an interesting variable to measure and report in the paper, but I don’t agree with it being used in this way in the formula. This is because the CC parameter already accounts for differences in the number of eggs produced – all else being equal, fewer eggs per egg string will result in fewer F1 larvae per egg string (this contributes to the observed effect on CC). Therefore, the vaccine effect on LE is already contained within the effect on CC, and multiplying the two parameters within the same formula is double-counting the FE effect and inflating the overall effect by a few %.

So I have to push for my original recommendation for the formula:

Vaccine efficacy (%) = 100 x [1-(FE x CC)].

Alternatively, the authors could include FE and CC, but in a different form where LE is the effect on number of eggs produced, and CC/LE is the effect on copepodid production from a given number of eggs (not egg strings). This avoids double-counting and gives the same result as 100 x [1-(FE x CC)], but explicitly shows the importance of LE by partitioning the vaccine effects on reproduction into (i) number of eggs per egg string, and (ii) number of larvae per egg:

Vaccine efficacy (%) = 100 x [1-(FE x LE x (CC/LE))]

For simplicity, CC/LE could be assigned to a new parameter called, for example, CE (copepodids per egg) to simplify the formula. The formula would then be:

Vaccine efficacy (%) = 100 x [1-(FE x LE x CE)].

Where FE is the effect on female survival to maturity, LE is the effect on fecundity of adult females (using egg string length as a proxy for fecundity), and CE is the effect on hatching and survival of F1 offspring to the copepodid stage.

7. PLOS authors have the option to publish the peer review history of their article (what does this mean?). If published, this will include your full peer review and any attached files.

Reviewer #1: No

Reviewer #2: No

---

## [Author Response · Author response to Decision Letter 1]

6 Sep 2020

Response to Reviewer

Reviewer #2: The authors have generally done a good job of revising the manuscript, and I have only two substantive concerns remaining (one of which still affects the overall vaccine efficacy estimate). I’m confident that both concerns can be addressed to my satisfaction.

The writing could still benefit from some copy editing, but I’ll leave this for the authors and/or journal to manage.

Well done to the authors for the effort they’ve put into this trial.

Authors response: Thanks a lot. We really appreciate reviewer’s effort for good self-explanatory advice and comments. 

1) Regarding lice and egg string loss during netting

Original comment:

L162: It is good that counting was done this way, but how were fish sampled? My experience is that most lice are actually lost during netting and anaesthesia rather than during counting. This paragraph should say how the sampling took place. For example, were the tanks sedated before sampling (this reduces the number of lice lost during netting), were fish netted or removed some other way (again, affects how many lice are lost), were the 10 fish per tank anaesthetised together in a bucket or separately, was the anaesthetic water checked for detached lice, etc.). This shouldn’t qualitatively change the vaccine effect, but it may change our understanding of infestation density across all three groups (especially because lice levels decreased markedly over time).

Authors’ response:

As per reviewer suggestion, the paragraph was modified accordingly in L170-178 as mentioned below: “Fish were taken out one at a time gently by handle fishing net from the stocking tanks and transferred in a bucket containing an overdose of anaesthetic water (0.01% benzocaine). Care was taken for the minimum loss of lice through netting. The net was simultaneously checked for the detached lice. One fish per bucket was anaesthetized before lice counting. To avoid counting error of detached lice due to anaesthesia and handling, counting of chalimus at 17 dpi, pre-adults at 28 dpi and adults at 50 dpi on individual parasitized fish were performed under water in a white tray. After lice counting from each fish/tray, the remaining water in the respective tray and bucket were checked for detached lice.” Moreover, we have experienced that after sedation of fish, lice is detached very quickly from the fish. Therefore, fish were taken out first one by one through handle net and then anaesthetised in a white bucket and simultaneously the net was checked for any detached lice on the net in anaesthetic water.

New comment:

This is a better description of the sampling protocol, although I’m still not convinced that the authors are able to prevent loss of any lice or egg strings during netting. As I noted in my original comment, I realise that this does not matter much for the measured vaccine efficacy because all groups received the same handling, but in a study that directly concerns infestation density and infestation persistence, likely sources of lice loss during sampling should be acknowledged. It is well known that netting salmon (and especially unsedated salmon) causes some adult lice and egg strings to be dislodged from abrasion by the mesh, and netting “gently” or “with care” doesn’t change this, especially if the fish is struggling. I would therefore like the authors to include a note either acknowledging this or explaining how they can be sure that none were lost. E.g. (i) maybe the net mesh was fine enough to capture lice and egg strings that might have been dislodged, (ii) maybe they are confident that they would see any dislodged egg strings in the tank after netting (detached lice would likely reattach to another host in the tank – something I’ve seen before), or (iii) acknowledge that some lice and egg strings could have been lost before the fish was transferred to the anaesthetic bucket.

A small point on wording: Change “handle fishing net” to your choice of “hand net”, “dip net” or “scoop net”

Authors Response: As suggested by the reviewer, we have acknowledged lice loss during sampling and added text accordingly in L171-174 of the revised manuscript in.

“All fish were treated the same and were handled gently. The hand nets were checked for detached lice and the net mesh was fine enough to capture lice if any lice would have fell off. However, we cannot exclude the possibility that some egg strings or lice would have been detached from the fish or lost in the tank during handling (netting).” 

As advised, we changed “handle fishing net” to “hand net”

2) Regarding the vaccine efficacy formula

Original comment:

L186: This formula seems problematic to me: “Vaccine efficacy (%) = 100 x [1-(NCh x NPA x NF x NM x FE x NE x LE x CC)]”. Unless I misunderstand, it looks like repeated counting of the same effect. For example, there is no need to include the immature stages here – losses of lice at all stages will accumulate and be ultimately represented by the difference in the density of adult females. You can then multiply that effect by effects on reproductive output per female. The study cited (Rodríguez-Mallon et al. 2015) used a similar formula but using different parameters, so does not support the usage here. I also disagree with the use of egg string number (NE), as this is more likely to be a sampling artefact than a biological effect (see also L256). Moreover, wouldn’t F1 copepodid counts also incorporate differences in egg string length (LE) if that variable is important? Unless the authors can explain why this isn’t repeated counting of the same effects, I recommend using a much simpler and more defensible formula to calculate an overall effect, representing how many female lice survive to maturity and how many larvae they produce: Vaccine efficacy (%) = 100 x [1-(FE x CC)]. Note, the CC parameter should be corrected for the number of egg strings collected per female (copepod count / number of egg strings).

Authors’ response:

Taking into account the reviewer suggestion, we assumed the proposed formula of vaccine efficacy but include the parameter LE as shown below:

Vaccine efficacy (%) = 100 x [1-(FE x LE x CC)].

LE: length of egg strings in vaccinated group/length of egg strings in control group. The length of egg strings are considered representative of how many eggs were produced, because it is almost impossible to count the eggs one by one.

CC: F1 generation copepodids count from vaccinated group/F1 generation copepodids count in control group. It is an indicator of how many larvae are produced from a fixed amount of egg strings. We think that the correction of CC is not valid since we placed the same number of egg strings per group (50 egg strings per group divided in 5 incubation chambers with 10 egg string in each one) in the incubation chambers for hatching.

With the new formula, vaccine efficacy was 58% for group 2 and 20% for group 3. This was corrected in the text.

New comment:

I agree with the authors that CC should not be corrected for the number of egg strings (it is now clear that the same number of egg strings were used per group).

However, the inclusion of LE still seems incorrect. I agree that LE is a fair proxy for number of eggs, and is an interesting variable to measure and report in the paper, but I don’t agree with it being used in this way in the formula. This is because the CC parameter already accounts for differences in the number of eggs produced – all else being equal, fewer eggs per egg string will result in fewer F1 larvae per egg string (this contributes to the observed effect on CC). Therefore, the vaccine effect on LE is already contained within the effect on CC, and multiplying the two parameters within the same formula is double-counting the FE effect and inflating the overall effect by a few %.

So I have to push for my original recommendation for the formula:

Vaccine efficacy (%) = 100 x [1-(FE x CC)].

Alternatively, the authors could include FE and CC, but in a different form where LE is the effect on number of eggs produced, and CC/LE is the effect on copepodid production from a given number of eggs (not egg strings). This avoids double-counting and gives the same result as 100 x [1-(FE x CC)], but explicitly shows the importance of LE by partitioning the vaccine effects on reproduction into (i) number of eggs per egg string, and (ii) number of larvae per egg:

Vaccine efficacy (%) = 100 x [1-(FE x LE x (CC/LE))]

For simplicity, CC/LE could be assigned to a new parameter called, for example, CE (copepodids per egg) to simplify the formula. The formula would then be:

Vaccine efficacy (%) = 100 x [1-(FE x LE x CE)].

Where FE is the effect on female survival to maturity, LE is the effect on fecundity of adult females (using egg string length as a proxy for fecundity), and CE is the effect on hatching and survival of F1 offspring to the copepodid stage.

Authors Response: First, we appreciate reviewer’s effort for good self-explanatory advices and comments. 

We agree with reviewer’s original recommended formula as well as the alternative formula “Vaccine efficacy (%) = 100 x [1-(FE x LE x CE)]”. Both the original and the alternative formula as suggested by the reviewer gave exactly the same value of vaccine efficacy. i.e 56% for group 2 and 25% for group 3. 

Here, we preferred to use the alternative formula as suggested by the reviewer, which accounts the effect on female fecundity as well as on hatching and survival of F1 offspring to the copepodid stage. The corrected vaccine efficacy formula and values are included in the revised manuscript. 

(Just to mention, there was a minor error in calculation for Gr 3 in the efficacy and in Table 2. This has been corrected in the revised version)

---

## [Editor Report · Decision Letter 2]

15 Sep 2020

Impact of a candidate vaccine on the dynamics of salmon lice (Lepeophtheirus salmonis) infestation and immune response in Atlantic salmon (Salmo salar L.)

PONE-D-20-16484R2

Dear Dr. Swain

We’re pleased to inform you that your manuscript has been judged scientifically suitable for publication and will be formally accepted for publication once it meets all outstanding technical requirements.

Kind regards,

Simon Clegg, PhD

Academic Editor

PLOS ONE

Additional Editor Comments:

Many thanks for resubmitting your manuscript to PLOS One

As you have addressed all the reviewer comments, and the manuscript reads well, I have recommended it for publication

You should hear from the Editorial Office soon

It was a pleasure working with you, and I wish you all the best for your future research

Hope you are keeping safe and well in these difficult times

Thanks

Simon

---

## [Editor Report · Acceptance letter]

23 Sep 2020

PONE-D-20-16484R2 

Impact of a candidate vaccine on the dynamics of salmon lice (*Lepeophtheirus*
*salmonis*) infestation and immune response in Atlantic salmon (*Salmo*
*salar* L.) 

Dear Dr. Swain:

I'm pleased to inform you that your manuscript has been deemed suitable for publication in PLOS ONE. Congratulations! Your manuscript is now with our production department. 

Kind regards, 

on behalf of

Dr. Simon Clegg 

Academic Editor

PLOS ONE